

**Secondary aerosol formation in marine Arctic environments: A model measurement**
**comparison at Ny-Ålesund**
Carlton Xavier[1], Metin Baykara[1,2], Robin Wollesen de Jonge[3], Barbara Altstädter[4], Petri
Clusius[1], Ville Vakkari[5.6], Roseline Thakur[1], Lisa Beck[1], Silvia Becagli[7], Mirko Severi[7], Rita
Traversi[7], Birgit Wehner[8], Mikko Sipilä[1], Markku Kulmala[1], Michael Boy[1], Pontus Roldin[3]
[1]Institute for Atmospheric and Earth Systems Research, University of Helsinki, P.O. Box 64,
00014 Helsinki, Finland
[2]Climate and Marine Sciences Department, Eurasia Institute of Earth Sciences, Istanbul
Technical University, Maslak 34469, Istanbul, Turkey
[3]Division of Nuclear Physics, Department of Physics, Lund University, P. O. Box 118 SE-221
00 Lund, Sweden
[4]Institute of Flight Guidance, Technische Universität Braunschweig, 38108 Braunschweig,
Germany
[5]Atmospheric Chemistry Research Group, Chemical Resource Beneficiation, North-West
University, Potchefstroom, South Africa
[6]Finnish Meteorological Institute, POBox 503, FI-00101 Helsinki, Finland
[7]Department of Chemistry, University of Florence, Sesto Fiorentino, 50019 Florence, Italy
[8]Institute of Tropospheric Research, 04318 Leipzig, Germany
**Correspondence:** Carlton Xavier (carlton.xavier@helsinki.fi) and Pontus Roldin
(pontus.roldin@nuclear.lu.se)
**Abstract**
In this study, we modeled the aerosol particle formation along air mass trajectories arriving at
the remote Arctic research stations Gruvebadet (67 m a.s.l) and Zeppelin (474 m a.s.l), Ny-
Ålesund during May 2018. The aim of this study was to improve our understanding of
processes governing secondary aerosol formation in remote Arctic marine environments. We
run the Lagrangian chemistry transport model ADCHEM, along air mass trajectories
generated with FLEXPART v10.4. The air masses arriving at Ny-Ålesund spend most of their
time over the open ice-free ocean. In order to capture the secondary aerosol formation from
the DMS emitted by phytoplankton on the ocean surface, we implemented a recently
developed comprehensive DMS and halogen multi-phase oxidation chemistry scheme,
coupled with the widely used Master Chemical Mechanism (MCM).
The modeled median particle number size distributions are in close agreement with the
observations in the marine influenced boundary layer at near sea surface Gruvebadet site.
However, while the model reproduces the accumulation mode particle number concentrations
at Zeppelin, it overestimates the Aitken mode particle number concentrations by a factor of





~5.5. We attribute this to the deficiency of the model to capture the complex orographic
effects on the boundary layer dynamics at Ny-Ålesund. The model also reproduces the
average vertical particle number concentration profiles within the boundary layer (0-600 m
a.s.l.) above Gruvebadet, as measured with Condensation Particle Counters (CPCs) on board
an Unmanned Aircraft Systems (UAS).
The model successfully reproduces the observed Hoppel minima, often seen in particle
number size distributions at Ny-Ålesund. The model also supports the previous experimental
findings that ion mediated $H_2SO_4$-$NH_3$ nucleation can explain the observed new particle
formation in the marine Arctic boundary layer in the vicinity of Ny-Ålesund. Precursors
resulting from gas and aqueous phase DMS chemistry contribute to the subsequent growth of
the secondary aerosols. The growth of particles is primarily driven via $H_2SO_4$ condensation
and formation of methane sulfonic acid (MSA) through the aqueous-phase ozonolysis of
methane sulfinic acid (MSIA) in cloud and deliquescent droplets.
**1. Introduction**
The Earth's radiation budget is influenced both directly and indirectly by aerosols, which
scatter and absorb the incoming short-wave radiation (direct effect) and serve as cloud
condensation nuclei (CCN, indirect effect), affecting both short and long-wave radiation
(Gantt et al., 2014; Oshima et al., 2020; Park et al., 2017; Scott et al., 2014). The Arctic
environments are susceptible to perturbations in the radiation balance, with some estimates
suggesting that, compared to the global average, the Arctic is warming at twice the rate, a
phenomenon termed as Arctic amplification (AMAP, 2011, 2017; Tunved et al., 2013). The
warming of the Arctic polar environment has accelerated sea ice loss, leading to a rapid
decline in the extent and duration of snow cover and increase in permafrost thaw (AMAP,
2011, 2017; Bengtsson et al., 2013).
The Arctic aerosol number concentration shows a pronounced seasonal variation, where the
late winter and early spring period is characterized by elevated accumulation mode aerosol
concentrations, accompanied by trace gases (mostly anthropogenic with long-range
transported trace elements such as sulfates, soot, and Peroxy Acyl Nitrates (PANs)). This
annually recurring phenomenon in late winter and spring is termed the Arctic Haze (Barrie,
1986; Lupi et al., 2016; Tunved et al., 2013). This contrasts with the summer period, when the
atmospheric new particle formation is observed at Arctic sites, most likely due to low
background aerosol concentrations, increased photo-chemistry and biological activity
(Engvall et al., 2008; Heintzenberg et al., 2017; Tunved et al., 2013).


The climate change driven Arctic sea ice loss has a profound impact on natural aerosol
production. Arrigo and van Dijken, 2015 found that decreasing and thinning of sea ice
increased the rates of phytoplankton net primary production by ~20% between the years 1998
and 2009. This can lead to an increase in the emissions of primary biogenic precursors such as
dimethyl sulfide (DMS), nitrogen volatiles (e.g. alkyl-amines) (Dall'Osto et al., 2017a;
Dallósto et al., 2017b) and biological iodine species (Cuevas et al., 2018). DMS is emitted
into the atmosphere via air-sea gas exchanges (Park et al., 2017; Uhlig et al., 2019), and
accounts for ~80% of global natural sulfur emissions (Kettle and Andreae, 2000; Uhlig et al.,
2019). Methane sulfonic acid (MSA) and sulphuric acid ($H_2SO_4$) is formed via DMS gas-
phase oxidation by OH and halogen species (Cl, Br) (Hoffmann et al., 2016; Kim et al., 2021;
Wollesen de Jonge et al., 2021). MSA and $H_2SO_4$, together with ammonia ($NH_3$) or amines,
act as precursors contributing to new particle formation (NPF) and subsequently to CCN
production, influencing cloud formation and radiative balance (Berndt et al., 2020; Dallósto et
al., 2017; Hoffmann et al., 2016; Kim et al., 2021). $NH_3$ plays a major role in particle
formation through stabilization of sulfuric acid clusters (Beck et al., 2021; Jokinen et al.,
2018; Olenius et al., 2013). Depending on local parameters such as ocean pH, salinity and
temperature, global oceans can act either as a source or sink of $NH_3$ (Paulot et al., 2015).
Apart from participating in cluster formation, $NH_3$ influences the pH of marine aerosols by
neutralizing the acid ($H_2SO_4$ and MSA) in the particles (Paulot et al., 2015). Though a few
potential sources of $NH_3$ are known, for example coastal sea bird colonies, pockets of open
water and melting sea ice in summertime Arctic, the magnitude of the emissions remain
uncertain (Dall'Osto et al., 2019; Riddick et al., 2012; Wentworth et al., 2016).
DMS oxidation chemistry has been under focus, but uncertainties in climate predictions
persist since the chemical transport models (CTMs) and global climate models (GCMs)
employ fixed MSA and $SO_2$ yields from gas-phase oxidation of DMS to calculate aerosol
formation (Hertel et al., 1994; Hoffmann et al., 2016; Kloster et al., 2006; Wollesen de Jonge
et al., 2021). Including a detailed multi-phase (aqueous-phase chemistry coupled with gas-
phase chemistry) DMS chemistry in numerical models can overcome these uncertainties
(Barnes et al., 2006; Campolongo et al., 1999). Reaction intermediates such as dimethyl
sulfoxide (DMSO), dimethyl sulfone ($DMSO_2$), methane sulfinic acid (MSIA) are water-
soluble, and experiments have shown that neglecting aqueous phase chemistry leads to either
an under-estimation of modeled MSA (Campolongo et al., 1999), or an over-estimation of
gaseous $SO_2$ compared to measured values (Hoffmann et al., 2016). For example, the
temperature dependent ratio of MSA/non-sea-salt $SO_4^{2-}$ (nss-$SO_4^{2-}$) is often used to estimate





the contribution of DMS to sulfate budget (Ayers et al., 1999; Barnes et al., 2006).
Campolongo et al., 1999 showed that modeling studies which included a multi-phase DMS
chemistry can bridge the gap between temperature-dependent observations and modeled
$MSA/nss\text{-}SO_4^{2-}$. Incorporating reactive halogens species over marine environments is crucial
in determining the DMS oxidation pathways to either $SO_2$ or MSA, the aging of marine
aerosols and the radiative properties of marine clouds (Hoffmann et al., 2016). Modeling
studies have shown that $Cl^-$ and $BrO^-$ radicals in the gas phase act as important DMS sinks
(Chen et al., 2018; Wollesen de Jonge et al., 2021), further underlining the role of halogen-
DMS chemistry in the marine boundary layer.
Recent DMS+OH oxidation experiments performed in the AURA chamber at Aarhus
University show that MSA dominates the secondary aerosol mass formation (Rosati et al.,
2021). Aerosol dynamics model simulations which intended to replicate the observations
during these AURA experiments, using the DMS gas-phase chemistry scheme from the
Master Chemical Mechanism, MCMv3.3.1, (Jenkin et al., 1997, 2015; Saunders et al., 2003),
substantially underestimates the particle mass and number concentrations and the $MSA:SO_4$
(Rosati et al., 2021, Wollesen de Jonge, 2021). Based on these findings, Wollesen de Jonge et
al. (2021) developed a new DMS multi-phase chemistry scheme based on MCM v3.3.1,
CAPRAM DMS module 1.0 (DM1.0) (Hoffmann et al., 2016), a subset of the multi-phase
halogen chemistry mechanism CAPRAM Halogen Module 2.0 (HM2.0) (Bräuer et al., 2013)
and new reactions leading to the formation of hydroperoxymethyl thioformate (HPMTF).
With the new DMS multi-phase chemistry mechanism, the aerosol dynamics model could
capture the observed particle number concentrations and secondary PM MSA and $SO_4^{2-}$
during DMS oxidation experiments performed at both dry and humid conditions at 0 ºC and
20 ºC in AURA.
In this work, we have implemented the above mentioned DMS multi-phase chemistry
mechanism into ADCHEM (see Methods section) and modeled the aerosol formation along
air mass trajectories arriving at Ny-Ålesund. We compared the model results with
observations from Zeppelin (78º56' N, 11º53' E, 474 m a.s.l) and Gruvebadet (78º92' N, 11º90'
E, 67 m a.s.l). These two sites represent remote marine Arctic conditions. Gruvebadet
represents ground-level concentrations as it is well within the boundary layer (BL). Zeppelin
on the other hand, is most often above the BL in winter months and sometimes below the BL
during spring and summer months (Traversi et al., 2020). This implies that Zeppelin is often
influenced by long range transport, and Gruvebadet by local short-range effects (Traversi et
al., 2020). This, demonstrates the complexity involved in capturing the atmospheric mixing





and secondary aerosol concentrations at Ny-Ålesund. The reason is that Svalbard has an
orographically complex terrain comprising of mountains, glaciers, fjords and flat lands that
introduce various micro-meteorological phenomena (Rader et al., 2021; Schemann and Ebell,

142   2020).


**2. Methods**
Using the combined multi-phase DMS chemistry mechanism by Wollesen de Jonge et al.,
(2021), MCMv3.3.1 and the monoterpene peroxy radical autoxidation mechanism (PRAM,
Roldin et al., 2019; Xavier et al., 2019) we simulated aerosol particle formation within the
marine boundary layer (MBL) upwind and at Ny-Ålesund between $1^{st}$ - $25^{h}$ May 2018, using
the Aerosol Dynamics, gas and particle-phase CHEMistry and radiative transfer model
ADCHEM (Öström et al., 2017; Roldin et al., 2011, 2019). We ran ADCHEM as a
Lagrangian model along the air mass trajectories arriving at Zeppelin every 3 hours during the
selected period (in total 200 trajectory simulations). FLEXPART v10.4 was used to calculate
the air mass trajectories and potential emission sensitivity fields (Pisso et al., 2019; Stohl et
al., 2005). The simulation results for the vertical distribution of newly formed aerosol (size <
12 nm) were validated against concurrent measurement data available from the ALADINA
(Application of Light-Weight Aircraft for Detecting in situ Aerosol) campaign, wherein a
UAS was used to investigate horizontal and vertical distribution of aerosol profiles in the
marine boundary layer (ABL) (Lampert et al., 2020). Additionally, modeled particle number
size distributions and $PM_{10}$ chemical compositions were compared to the available measured
particle number size distributions and $PM_{10}$ filter samples at both Gruvebadet and Zeppelin
measurement stations.
**2.1 Air mass trajectories and potential emission sensitivity fields**
We employed the Lagrangian particle dispersion model FLEXible PARTicle
(FLEXPARTv10.4) to assess the emission sensitivities or "footprints" of air-masses origin
arriving at Zeppelin during the simulation period. FLEXPART is a stochastic model used to
compute dispersion of hypothetical particles, based on mean, turbulent and diffusive flows
which can be run backwards in time to estimate air mass history at a site (Pisso et al., 2019).
European Center for Medium-Range Weather Forecasts (ECMWF) ERA5 reanalysis
meteorology with 137 height levels, 1-hour temporal and 0.5° x 0.5° spatial resolution, was
used as an input to FLEXPART (*ERA5 hourly data on single levels from 1979 to present.*





*Copernicus Climate Change Service (C3S) Climate Data Store (CDS). last access 30th April*
*2021, 10.24381/cds.adbb2d47, ERA5 hourly data on pressure levels from 1979 to present.*
*Copernicus Climate Change Service (C3S) Climate Data Store (CDS). last access 30th April*
*2021, 10.24381/cds.bd0915c6*). The air-mass history was simulated 7-day backwards in time
and arriving at Zeppelin (474 m a.s.l) every 3 hours (at 00:00, 03:00, 06:00, 09:00, 12:00,
15:00, 15:00 and 21:00 UTC) for the entire simulation period (1$^{st}$ - 25$^{th}$ May 2018).
FLEXPART calculated normalized emission sensitivity fields were combined with oceanic
emissions (DMS, dibromomethane, tribromomethane, iodomethane), $NH_3$ from sea-bird
colonies and anthropogenic emissions ($NH_3$, $SO_2$, CO, $NO_x$) derived from global inventories
(see section 2.2). This was done to obtain representative emissions that consider the complete
emsission source regions along the trajectories, upwind of the measurement station.
Additional meteorological parameters such as temperature, pressure, sea surface temperature,
specific humidity and cloud liquid water content from ERA5 reanalysis dataset were extracted
along the trajectories and provided as inputs to ADCHEM.

**2.2 Gas and primary particle emissions**
Emissions of gas-phase biogenic volatile organic compounds (VOCs) α-pinene, β-pinene Δ3-
carene, limonene, isoprene and β-caryophyllene were modeled with a 1 - dimensional version
of MEGAN v2.04 (Model of Emissions of Gases and Aerosols from Nature 2.04) (Guenther
et al., 2006). Gas-phase emissions of marine halogens such as tribromomethane ($CHBr_3$),
dibromomethane ($CH_2Br_2$), iodomethane ($CH_3I$) were retrieved from CAMS-OCE Global
oceanic emissions (CAMS-GLOB-OCE) which are available as daily means with a spatial
resolution of 0.5°x0.5° (Granier et al., 2019; Ziska et al., 2013). CAMS-GLOB-OCE also
provides gas-phase DMS emissions with the same temporal and spatial resolution (Granier et
al., 2019) calculated with the air-sea flux parameterization and emission fluxes described in
(Lana et al., 2011; Nightingale et al., 2000). $NH_3$ emissions from seabird colonies were
acquired from a global emission inventory (Riddick et al., 2012). To account for additional
$NH_3$ fluxes from the open ocean, we used an estimated sea surface equilibrium $NH_{3(g)}$
saturation concentration of 0.5 nmol/m$^3$ (12.2 ppt at standard temperature and pressure (STP))
which approximately correspond to a surface ocean ammonium concentration of 0.125
mmol/m$^3$ (or ~3ppb, calculated based on equation 3 & 4 from Wentworth et al., 2016) at a sea
surface temperature of +2 °C. The sea surface temperature for the study period varied between
0 °C-14 °C along the trajectories. The estimated surface ocean ammonium concentrations is in



close agreement with the concentration estimated by the global ocean biogeochemical model
COBALT (Stock et al., 2014) in the North Atlantic ocean, but up to a factor of ~5 higher than
the concentrations simulated with other ocean biogeochemical models and/or model setups
(Paulot et al., 2015). Therefore, we performed model sensitivity runs with a sea surface
equilibrium $NH_{3(g)}$ concentration of 0.1 and 1 nmol/m³. The $NH_{3(g)}$ equilibrium saturation
concentrations represent the ambient surface gas-phase concentration at which the air-sea flux
changes direction, with a net downward flux from air to sea if the ambient $NH_{3(g)}$ exceeds the
equilibrium gas concentrations and vice versa (Wentworth et al., 2016). For the anthropogenic
trace gas and primary particle emissions, we used the CAMS-GLOB-ANT v2.1 inventory,
with a spatial resolution of 0.1°x0.1° (Granier et al., 2019).
In this work, we used the sea surface temperature (SST) and wind speed dependent sea-spray
aerosol (SSA) emission parameterization by Sofiev et al. (2011), (further referred to as
Sofiev11). Sofiev11 used a modified source function based on the parameterization of
Monahan et al. (1986) and experiments by Mårtensson et al. (2003) and SEAS campaign by
Clarke et al. (2006). The modified source function in Sofiev11 provides extrapolated SSA
emissions between size ranges of 10 nm-10 μm, with appropriate correction functions
employed for SST deviating from 298.15 K (Sofiev et al., 2011). Sofiev11 SSA
parameterization shows that with increasing temperatures, emission flux for larger particles
increases while the emission fluxes for smaller particles decreases (Barthel et al., 2019;
Sofiev et al., 2011). We performed sensitivity tests using the temperature and wind speed
dependent SSA parameterization by Salter et al., (2015), (further referred to as Salter15).
Both the Salter15 and Sofiev11 are valid between 10 nm-10 μm. Model simulation
comparisons between Sofiev11 and Salter15 have shown that the SSA parameterization from
Sofiev11 has a stronger temperature dependence and higher particle number concentration
emissions in the Aitken mode but result in lower $PM_{10}$ emissions at temperatures below 25 ºC
(Barthel et al., 2019).

## 2.2 ADCHEM

For this study, ADCHEM was employed as a 1 - dimensional column model with 40
logarithmically vertical layers, extending up to ~2600m. The model time step used for
simulations was 30 seconds. The vertical atmospheric turbulent diffusion was solved using a
modified Grisogono turbulent diffusivity scheme (Jeričević et al., 2010; Öström et al., 2017;
Roldin et al., 2019). The ADCHEM aerosol module includes new particle formation,





Brownian coagulation, condensation and evaporation of particles, and finally the dry and wet
deposition of both particles and gases. The particle number size distributions were represented
using 100 size bins ranging from 1.07 nm to 10 μm dry diameter. Clouds were assumed to be
present in the model grid cells when the bulk liquid water content (LWC, extracted along the
trajectory from ERA5 datasets) was greater than 0.01 g m$^{-3}$. As a default, we used a constant
cloud supersaturation ($S$) of 0.5% and the particles were activated into cloud droplets, if the
calculated water vapor supersaturation above the particle surface ($S_c$, calculated using Köhler
theory) was smaller than $S$. During the cloud processing, each activated cloud droplet was
assumed to take up an equal amount of liquid water corresponding to the total bulk LWC
divided by the calculated number concentration of activated cloud droplets. The gas-liquid
droplet mass transfer and dissolution of 50 species in total, including HCl, HNO$_3$, H$_2$SO$_4$,
NH$_3$, HIO$_3$, H$_2$O$_2$, O$_3$, OH, BrO, NO$_3$, DMSO, MSIA, MSA and HPMTF and their irreversible
reactions in the interstitial and activated cloud droplets are treated by the multi-phase
chemistry mechanism (see Wollesen de Jonge et al. (2021) for details). The kinetic pre-
processor (KPP) (Damian et al., 2002) was used to generate the multi-phase chemistry
mechanism used in this study.
Recent observations of NPF at Ny-Ålesund have confirmed the importance of ion-mediated
H$_2$SO$_4$-NH$_3$ nucleation in spring with MSA and H$_2$SO$_4$ condensation contributing to the
subsequent growth of particles (Beck et al., 2021; Lee et al., 2020). In this work, the
Atmosphere Cluster Dynamics Code (ACDC) (McGrath et al., 2012; Olenius et al., 2013) was
coupled with ADCHEM (Roldin et al., 2019). ACDC was used to model NPF, which
involved H$_2$SO$_4$ clustering with NH$_3$ via both neutral and ion-induced pathways with an
ionization rate of 1.7 cm$^{-3}$s$^{-1}$. ACDC was used to solve the evolution of molecular H$_2$SO$_4$-NH$_3$
clusters by considering the loss of clusters by collisions, evaporation or coagulation
scavenging onto larger aerosol particles. At each time step, the flux of clusters (up to ~ 5
H$_2$SO$_4$ and 5 NH$_3$ each) growing out of the ACDC molecule-cluster domain represents the
NPF rate. These newly formed clusters are assigned to the corresponding smallest particle
size bin at 1.07 nm in diameter in ADCHEM, which then simulates the condensational growth
of particles and losses due to evaporation, coagulation, and wet and dry deposition.
For all simulations, we used model output from the closest height levels which can represent
Gruvebadet (model height of 73.5 m a.s.l) and Zeppelin (model height of 486.0 m a.s.l).
**Sensitivity Tests**
Alongside the main ADCHEM simulations, *BaseCase,* we performed nine complementary



scenario runs to assess the impact of different processes on the modeled aerosol
concentrations. We performed simulations without aerosol in-cloud processing (*Cloudoff*), to
check the impact of in-cloud processing on the growth of aerosols. We investigated the effect
of higher $PM_{10}$ particle emissions on the chemical composition of secondary aerosols, using
the sea-spray emission parameterization based on Salter et al., 2015 (*SalterSSA*). Simulations
were conducted to assess the impact of lower and higher ammonia sources over the open
ocean (*LowNH₃, HighNH₃*). A sensitivity test without precipitation (*NoPrecip*) was performed
to test the influence of precipitation on number concentration and particle composition. Since
cloud supersaturation is critical to the activation of particles and is highly uncertain, we
performed two simulations with low and high cloud supersaturation (*S=0.2%, SSat=0.2* and
*S=0.8%, Ssat=0.8*) to test its impact on the modelled particle distributions. We performed a
simulation without new particle formation (*NPFoff*), and finally one simulation without the
dissolution and irreversible aqueous chemistry of the intermediate DMS oxidation products,
$SO_2$ and halogens (*woDissolution*), implying that MSA, $H_2SO_4$ and $HIO_3$ is only formed in the
gas-phase. Table 1. summarizes the setup for different model sensitivity test.

**Table 1.** Model sensitivity tests performed alongside the main *BaseCase* simulations to test
the effect of different parameters on secondary aerosol formation. These sensitivity tests focus
on the role of in-cloud processing and aqueous phase chemistry, the $NH_3$ emissions from open
ocean, SSA parameterization and cloud supersaturation. The sea surface equilibrium $NH_{3(g)}$
concentrations in ppt are provided in the brackets.

| Simulation | In-cloud Processing | $NH_{3(equilibrium)}$ (nmol/m³, ppt ) | SSA parameterization | Precipitation |
|---|---|---|---|---|
| *BaseCase* | On | 0.5  (12.2) | Sofiev11 | On |
| *SalterSSA* | On | 0.5 (12.2) | Salter 15 | On |



| *Cloudoff* | Off | 0.5 (12.2) | Sofiev11 | On |
|---|---|---|---|---|
| *LowNH₃,* *HighNH₃* | On | 0.1 (2.4) 1 (24) | Sofiev11 | On |
| *NoPrecip* | On | 0.5 (12.2) | Sofiev11 | Off |
| *SSat0.8, SSat0.2* | On | 0.5 (12.2) | Sofiev11 | On |
| *NPFoff* | On | 0.5 (12.2) | Sofiev11 | On |
| *WoDissolution* | On, but no dissolution and irreversible chemistry of intermediate DMS oxidation products | 0.5 (12.2) | Sofiev11 | On |


**2.3 Measurements**
We utilized comprehensive measurements from the Ny-Ålesund sites, Zeppelin and
Gruvebadet during the period of 1ˢᵗ - 25ᵗʰ May 2018. Since 2017, the atmospheric observatory
at Gruvebadet, which is located about 700 m southwest of Ny-Ålesund village at almost sea
level (67 m.s.l), hosted Neutral cluster and Air Ion Spectrometer (NAIS, Manninen et al.,
2010; Mirme and Mirme, 2013) for semi-permanent measurements. Here we use NAIS
measured number size distribution of naturally charged (ions) in diameter size ranges between
0.8 nm-40 nm and neutral particles in the size range of 2.5 nm-42 nm, with a temporal
resolution of two seconds.





During the measurement period, a scanning mobility particle sizer (SMPS), was operated to
measure particle number size distribution in the diameter size range of 10 - 470 nm at
Zeppelin. Concurrent SMPS data (TSI 3034, 54 channels) with diameter size ranging from 10
to 470 nm from Gruvebadet were also available (Dall'osto et al., 2019; Moroni et al., 2020),
thus, enabling us to compare the modeled particle number size distribution with the measured
size distributions at both measurement stations. Daily resolution continuous aerosol samples
with $PM_{10}$ cutoff were collected at Gruvebadet using a Tecore Skypost low-volume sampler
(Amore et al., 2022). The detection limit for $Na^+$ was 0.0001 µg m$^{-3}$ and 0.0002 µg m$^{-3}$ for $Cl^-$,
$NH_4^+$ and $SO_4^{2-}$. Since the field blank medians at Gruvebadet were less than 1 percentile of
sampled values, the field blanks were not subtracted from the sampled values (Amore et al.,

311    2022).

Vertical particle number concentration profiles were obtained using UAS ALADINA (Bärfuss
et al., 2018; Lampert et al., 2020), which was operated during the simulation period.
ALADINA were operated up to a height of 850 m a.s.l., thus can be used for a potential
closure between the two different research sites of Gruvebadet and Zeppelin. ALADINA is
equipped with two condensation particle counters (CPCs  Model 3007, TSI Inc., St. Paul,
MN, USA), measuring in the size ranges of 3 nm - 2 µm (CPC1) and ~12 nm - 2 µm (CPC2)
(Lampert et al., 2020; Petäjä et al., 2020) . The difference between CPC1 and CPC2 provides
an estimate of particle number concentrations in the size of 3 - 12 nm ($PN_{3-12}$), which was
used as an indicator of NPF. Alongside the CPCs, a host of other instruments measuring
meteorological parameters were operated in unison, the description of which can be found in
Bärfuss et al., (2018) and Lampert et al.,(2020).
**Evaluating temporal aspects of model performance**
The modeled $PM_{10}$ inorganic chemical composition was evaluated against the measured $PM_{10}$
inorganic chemical composition using statistical estimates such as, normalized mean bias
(NMB), Pearson correlation coefficient (r), root mean squared error (RMSE) and fraction of
predictions within a factor of 2 of the observed values (FAC2). These tests were used to
evaluate modeled values ($M_i$) against observation values ($O_i$) at both the measurement sites.
Pearson correlation coefficient was calculated using the formula:
$$r = \frac{1}{n} \sum_{i=1}^{n} \frac{(O_i - \bar{O})}{\sigma_O} \frac{(M_i - \bar{M})}{\sigma_M} \qquad \text{(Eq 1)}$$

Where $\sigma_o$ and $\sigma_M$ are standard deviations of the observed and modeled values, respectively.



Normalized mean bias (NMB) indicates if the predictions are over or underestimating the
observed values, with the factor representing the under or over estimation. NMB was
calculated using Eq. 2:

$$NMB = \frac{\sum_{i=1}^{n}(M_i - O_i)}{\sum_{i=1}^{n} O_i}$$

(Eq 2)

Root mean squared error (RMSE) was calculated using Eq. 3:
$$RMSE = \sqrt{\sum_{i=1}^{n} \frac{(M_i - O_i)^2}{n}}$$ (Eq 3)

FAC2 is a robust metric defined as the percentage of predictions which are within a factor of
2 of the observed values (Eq. 4):

$$Fac2 = 0.5 \leq \frac{M}{O} \leq 2.0$$

(Eq 4)
**3. Results and Discussion**
In the following sections, we analyze and evaluate the model results against comprehensive
measurements in Ny-Ålesund. In sub-section 3.1, we focus on the particle number size
distributions at both sites, followed by gas-phase concentrations and $PM_{10}$ inorganic chemical
composition (sub-section 3.2) and the vertical nano-particle concentration profiles (sub-
section 3.3). Finally, in sub-section 3.4, we analyze the results from the model sensitivity
tests.

**3.1 Particle number size distributions**
Figure 1(a) and (b) show the observed and predicted particle number size distributions at
Gruvebadet for the *BaseCase* simulation. Figure 1(a) includes SMPS observations starting
from 10 to 470 nm and NAIS observations for neutral particles in the range 2.5 nm-10 nm
(boundary marked by the black line) since NAIS data below 2.5 nm cannot be relied upon,
owing to the presence of corona generated ions (Jayaratne et al., 2017; Manninen et al., 2011,

358 2016).

In the *BaseCase* simulations, ADCHEM reproduces the general trends of the observed
particle number concentrations. For example, the model captures particle formation on 2[nd]



May followed by an increasing number of Aitken and accumulation mode particles during the
days of $3^{rd}$ - $4^{th}$ May, which is the result of more polluted air masses arriving at Ny-Ålesund
from the European continent (Figure S1). Similarly, the model reproduces the particle
formation on the $20^{th}$ of May, specifically in the size ranges 2-8 nm, but overestimates the
Aitken mode and accumulation mode particle concentration on the $21^{st}$ of May. In general, the
model predicts the formation of new particles with reasonable accuracy during the selected
period. However, the model tends to underestimate the nucleation mode particle number
concentrations between 10-25 nm ($PN_{10\text{-}25\,nm}$) around noon, and overestimate the
concentrations during the morning and evening (Figure 2a). The model and measurements
show an apparent time delay in the formation of new particles larger than 10 nm. While the
measurements show a peak at 11 am the simulated $PN_{10\text{-}25\,nm}$ shows a maximum at 3 am and 6
pm. The modeled $PN_{10\text{-}25\,nm}$ maximum around 6 pm is likely a result of the formation of new
particles around noon, which grow to >10 nm in diameter during the afternoon and evening
by condensation of $H_2SO_4$. The predicted Aitken ($PN_{25\text{-}100\,nm}$) and accumulation mode particle
concentrations ($PN_{>100\,nm}$) which form few days upwind of the station are overall, in good
agreement with the measurements, which show a minor diurnal trend (Figure 2b-c). The
measurements indicate that at Gruvebadet, $PN_{10\text{-}25\,nm}$ contributes the most significant fraction
of measured total number concentrations with 45.3%, while $PN_{25\text{-}100\,nm}$ and $PN_{100\text{-}470\,nm}$
contribute 30.5% and 23.94% respectively. However, the simulations predict greater
contribution of Aitken mode (~53.85%) to total number concentration, with the $PN_{10\text{-}25\,nm}$ and
$PN_{100\text{-}470\,nm}$ accounting for ~ 36.58% and 9.57% respectively (Figure S2).
Figure 2 shows the measured size distribution in panel (a) and simulated size distribution in
panel (b) for Zeppelin. At Zeppelin, the model overestimates the number concentration in
nucleation and Aitken modes (also cf. Figure S3, supplementary). The particle number size
distribution measurements at Zeppelin indicate that the relative contribution of the three
modes (nucleation, Aitken and accumulation) varies to some extent when compared to
Gruvebadet. Measurements show that at Zeppelin, $PN_{10\text{-}25\,nm}$ contributes ~33.46%, $PN_{25\text{-}100\,nm}$
46.43% and $PN_{>100\,nm}$ 20.11% to the total particle number concentrations. The model predicts
lower relative contribution of $PN_{10\text{-}25\,nm}$ (26.94%), and a greater contribution of $PN_{25\text{-}100\,nm}$
(63.44%) to the total simulated particle number concentrations. The diurnal trends at Zeppelin
agree well with earlier measurements conducted at Zeppelin in spring by Ström et al. (2009).
Additionally, the measured diurnal pattern at Zeppelin varies in comparison to Gruvebadet. At
Zeppelin, the $PN_{10\text{-}25\,nm}$ concentrations peak in the afternoon and evening. The modeled
$PN_{10\text{-}25\,nm}$ shows only a weak diurnal trend. It should be noted that the measurements show a



time delay of around 3 hours in the peak $PN_{10-25\,nm}$ concentrations at the two sites (Figure S2
and S3). This is possibly be a result of vertical mixing and dilution effects modulating the
observed particle number concentrations at sites situated at different altitudes, similar to
observation made at Zeppelin and Corbel by Ström et al. (2009).
ADCHEM considers the formation of new particles via both the ion-mediated and neutral
$H_2SO_4$-$NH_3$ clustering pathways. Beck et al (2021) observed dominant contribution of
negative $H_2SO_4$-$NH_3$ clusters to secondary particle formation in May 2017 at Ny-Ålesund,
with $HIO_3$ playing a small role in the initial particle formation. However, the discrepancy in
the modeled and observed diurnal trends of $PN_{10-25\,nm}$ could indicate that there are other
sources or vapors that might potentially contribute to the particle formation. Other possible
NPF mechanism may involve amines (Olenius et al., 2013) and pure biogenic highly oxidized
molecule (HOM) (neutral and ion induced) nucleation (Kirkby et al., 2016). We speculate that
the exclusion of these other mechanisms ($HIO_3$, $H_2SO_4$-amines and HOM driven particle
formation) might result in the discrepancies in the modeled and observed particle number
concentration diurnal trends. $HIO_3$ induced particle formation could, e.g. play an important
role if the air masses upwind of Ny-Ålesund traverse over the sea-ice covered regions
(Baccarini et al., 2020; Beck et al., 2021).

Gruvebadet (Simulation: *BaseCase*)

**Figure 1.** Particle number size distribution at Gruvebadet for *BaseCase*. The panel **(a)** shows the measurement data for the period 1-25th May from SMPS (10 nm-470 nm) and NAIS (2.5 - 10 nm) and the panel **(b)** provides the modeled particle size distribution. The black line at 10 nm denotes the boundary above which SMPS data starts and NAIS data ends. The abscissa indicates the time for the entire simulated duration. The ordinate in Figure 1 for both panels (a) and (b) indicates the particle diameter ($D_p$, nm).

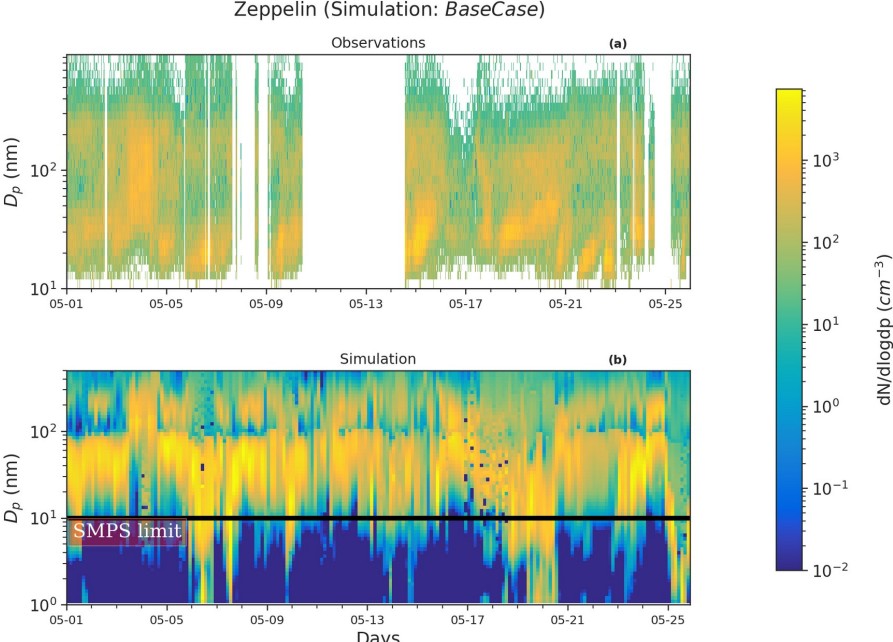

**Figure 2.** Particle size distribution at Zeppelin. The panel **(a)** shows the measurement data for
the period 1-25th May from SMPS and the panel **(b)** provides the simulated particle size
distribution for the *BaseCase* simulations. The abscissa and ordinates are similar to Figure 1.

Figure 3 presents the median particle number size distribution for the *BaseCase* simulation at
both Zeppelin and Gruvebadet, with respective 25th and 75th percentiles, for the entire selected
period. At Gruvebadet, the modeled and measured median particle number size distributions
are in reasonable agreement for both Aitken and accumulation mode. However, the model
over predicts the median Aitken mode concentrations at Zeppelin by a factor ~ 5.5. The
modeled Aitken mode peak at both measurement sites is ~50 nm, while the measured Aitken
mode peak is ~30 nm. Though the modeled accumulation mode peak is at a larger size (~150
nm), compared to the measured accumulation mode peak (~110 nm), the predicted values are
in good agreement with the monthly averaged accumulation mode peak location measured at
Zeppelin in earlier studies (~160-170 nm, Dall'Osto et al., 2019).

The discrepancy between the modeled and measured particle concentrations at Zeppelin can



be caused by the underlying complexity of modeling the boundary layer dynamics at an
elevated site, such as Zeppelin. The vertical mixing of aerosols along the up-slope or down-
slope of a mountain site is difficult, if not impossible for a 1- dimensional column model,
since it is unable to capture the topographical influence on locally varying wind speeds or
latent and sensible heat fluxes (Mikkola, 2020; Wainwright et al., 2012).

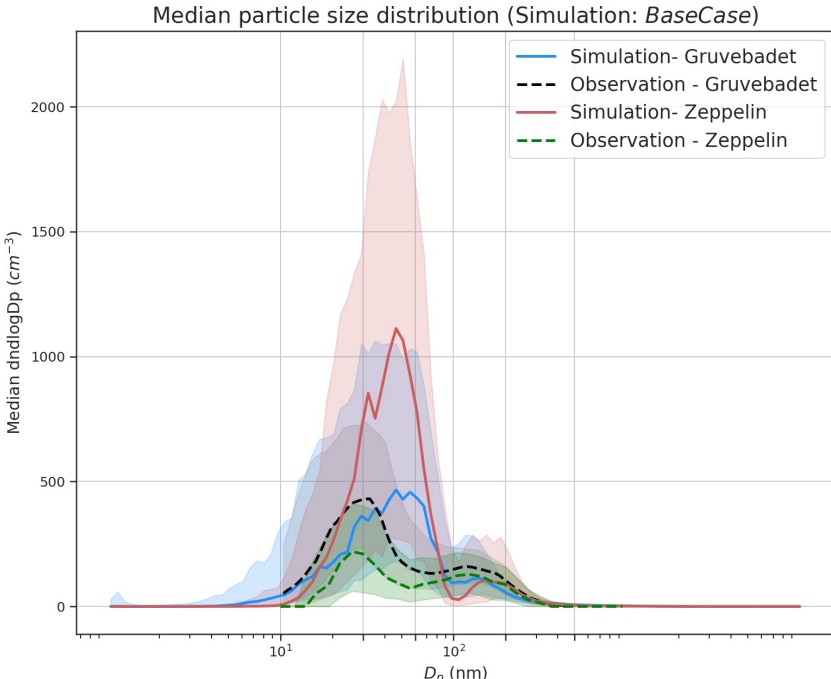

**Figure 3.** Median particle number size distribution at Gruvebadet and Zeppelin for both
modeled (*BaseCase* simulations) and measured values. The shaded areas indicate the 25th and
75th percentiles for both model and measured median particle number size distribution. At
Zeppelin, the simulated median size distribution is calculated for periods only when SMPS
data were available.

Another detectable feature in the median particle number size distribution is the diameter of
the Hoppel minimum (Hoppel et al., 1985, 1986), and the role of in-cloud processing in
forming this minimum. A Hoppel minimum is often observed in marine air masses (Fossum
et al., 2018; Tunved et al., 2013; Zheng et al., 2018) and is attributed to in-cloud processing of



aerosols, with chemical processing (e.g., sulfate production via oxidation of dissolved $SO_2$)
(Feingold and Kreidenweis, 2000; Hoppel et al., 1986), and coalescence of droplets playing a
key role (Flossmann and Wobrock, 2019; Hoppel et al., 1986; Hoppel and Frick, 1990; Noble
and Hudson, 2013). It has been estimated that, on average, aerosols take part in about 10 non-
precipitating cloud cycles before it is removed from the atmosphere by wet scavenging
(Hoose et al., 2008; Hoppel et al., 1986; Rosenfeld et al., 2014). These non-precipitating
cloud cycles facilitate the formation of hygroscopic accumulation mode particles, with low
critical supersaturation ($S_c$) that readily activates to cloud droplets during subsequent cloud
cycles, thus growing to larger sizes. This is because the activated particles undergo chemical
processing, gas-to-particle conversions, coalescence and coagulation with other interstitial
particles. Upon evaporation of water, the emerging dry particles have a larger size and lower
$S_c$, leading to a minimum being formed between the un-activated and activated cloud droplets
(Herenz et al., 2018; Hudson et al., 2015; Noble and Hudson, 2013). The diameter at which
the Hoppel minimum is observed varies depending on the cloud supersaturation and particle
composition (Hoppel et al., 1986; Hudson et al., 2015), with Hoppel minima sizes observed in
ranges from 60 nm at Zeppelin Ny-Ålesund to around 90 nm at Tuktoyatuk, Canada (Herenz
et al., 2018; Tunved et al., 2013).
The median particle number size distribution in Figure 3 shows that at both stations, the
measured Hoppel minima is around ~ 60 nm, while the simulated Hoppel minima are around
the size of ~ 100 nm at both sites. This difference in location of Hoppel minima can be
attributed to the assumed value of $S$=0.5% in the model. The value of $S$ used in the model lies
in the range of typical marine stratocumulus clouds, which can vary between 0.1 - 1%
(Fossum et al., 2018; Quinn et al., 2017).
**3.2 Gas and particle-phase chemical composition of important precursors**
Figure 4 shows the simulated gas-phase precursor and main DMS oxidation product
concentrations including $H_2SO_4$, MSA, MSIA, HPMTF, $SO_2$ and DMSO, for the entire period
at the height levels representing Gruvebadet (G1 and G2), and Zeppelin (Z1 and Z2). The $SO_2$
gas-phase concentrations are in the order of $10^6$-$10^9$ # $cm^{-3}$ (with monthly mean values 1.7 x
$10^8$ # $cm^{-3}$), which is a factor of 2.3 higher than the average concentrations measured for
spring 7.6 x$10^7$ # $cm^{-3}$ by (Lee et al., 2020) at Zeppelin. The monthly mean simulated $H_2SO_4$
gas phase concentrations (6.8 x $10^5$ # $cm^{-3}$) also agree well with the estimated $H_2SO_4$ proxy
(Eq. S1, supplementary) spring average values of 7.5 x $10^5$ # $cm^{-3}$ (Lee et al., 2020) at
Zeppelin. Measurements of $H_2SO_4$ at Gruvebadet from May 2017 indicate monthly mean





concentrations around ~$10^6$ # cm$^{-3}$ (Beck et al., 2021). The modeled H$_2$SO$_4$ concentrations at
Gruvebadet are 3 x $10^6$ # cm$^{-3}$, implying a reasonably good model performance in predicting
gaseous precursor concentrations. Simulated gas concentrations of MSA ($10^5$-$10^8$ # cm$^{-3}$ also
agrees well with the measurements made at Gruvebadet in May 2017 by (Beck et al., 2021),
wherein they measured daily averages of MSA gas concentrations in the order of $10^7$ # cm$^{-3}$.
The low modeled values of MSA and DMSO gas phase concentrations at the height
representing Zeppelin (e.g. between 15/05 - 17/05) coincide with the period where the
planetary boundary layer height (PBLH) is below the altitude of Zeppelin station (cf. Figure
S4 supplementary). Overall, we can conclude that the modeled precursor gas concentrations at
the two measurement sites are, in general, good agreement with earlier measurements at the
two sites.

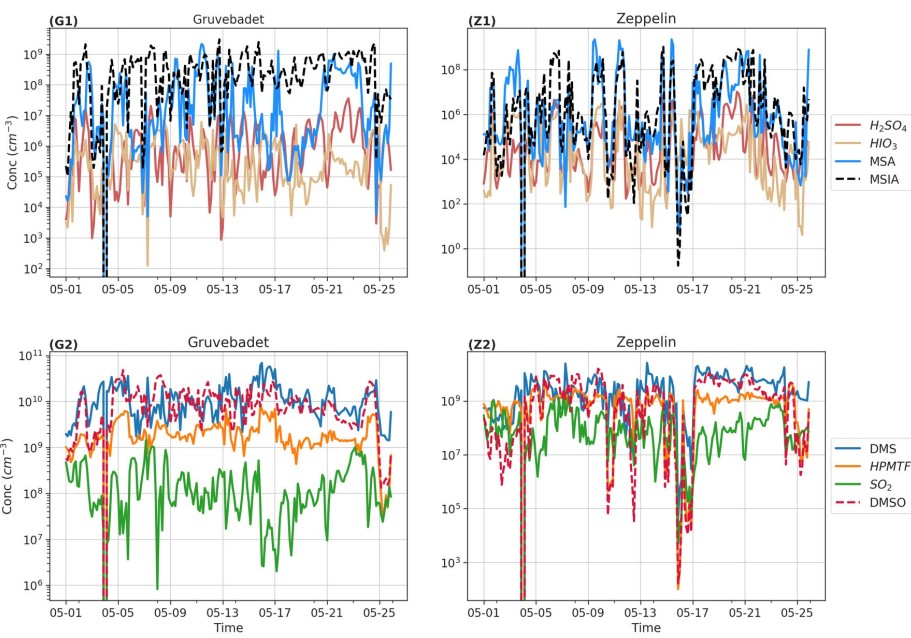

**Figure 4.** Gas-phase concentrations for the *BaseCase* simulations. The upper two panels **(G1)**
and **(Z1)** show the gas-phase concentrations at Gruvebadet and Zeppelin respectively, for
compounds H$_2$SO$_4$ (red), HIO$_3$ (gold), MSA (cyan), MSIA (dotted black) and the lower panels
**(G2)** and **(Z2)** show the gas-phase concentrations for DMS (blue), HPMTF (orange), SO$_2$




(green) and DMSO (dotted red). Note the different ordinate scales.

Figure 5 (a) shows the simulated median mass size distribution of compounds $Cl^-$, $Na^+$, MSA,
$SO_4^{2-}$, $NH_4^+$, and $NO_3^-$ for the *BaseCase* runs in the lowest model layer. Figure 5(a) indicates
that the nucleation mode particles are composed mainly of $SO_4^{2-}$ and $NH_4^+$, while MSA, $Cl^-$
and $Na^+$ dominate PM for larger particles. The observed and modeled high $MSA_{(g)}$
concentrations in comparison to $H_2SO_{4(g)}$ at Ny-Ålesund is not reflected in the respective
vapor contribution to the nano-particle growth. This is because, in contrast to $H_2SO_4$, MSA is
not a non-volatile condensable compound. The gas-to-particle partitioning of MSA requires
co-condensation and dissolution of $(NH_3)$ (Hodshire et al., 2019) or the existence of cations
such as $Na^+$ which decreases the particle acidity ($[H^+]$). Figure 5(b) shows the relative mass
fraction of the above-mentioned compounds to PM at different sizes. $SO_4^{2-}$ and $NH_4^+$
dominate the mass for particles in the nucleation and Aitken mode. $SO_4^{2-}$ contributes ~74%
and ~71% to nucleation and Aitken mode PM, with its contribution decreasing for
accumulation (100 nm-1 μm) and coarse (>1 μm) mode PM (~6% and 3.36% respectively)
(Table 2). $NH_4^+$ contribution follows a similar trend, as $SO_4^{2-}$, with 12.34% and 6.95%
contribution to nucleation and Aitken mode PM, but insignificant for accumulation and coarse
mode PM (Table 2). The loss of primary sea spray aerosols due to wet scavenging promoted
the growth of secondary aerosol particles in the nucleation and Aitken mode by $NH_4^+$ and
$SO_4^{2-}$ as seen in Figure 5 (b). $Na^+$ (~32.9%), $Cl^-$ (~39.5%) and MSA (20.45%) are the
dominant contributors to accumulation and coarse mode PM. In the *BaseCase* simulations,
gas-phase $SO_2$ dissolves in the cloud droplets, and is oxidized by $H_2O_2$ into $SO_4^{2-}$ (Wollesen
de Jonge et al., 2021). Previous modeling studies have shown that a very small fraction of
MSA is formed in the gas phase. Instead, most MSA is formed via ozonolysis of MSIA in the
aqueous phase (Hoffmann et al., 2016; Wollesen de Jonge et al., 2021). It should be noted that
$HIO_3$ and $NO_3^-$ have an insignificant contribution to total $PM_{10}$, amounting to ~0.05% and
0.17% respectively.

**Table 2**: The table shows simulated fractional contribution of different compounds to total
PM in different size regimes of nucleation (total $PM_{<25nm}$) , Aitken (total $PM_{25 - 100 nm}$) and
accumulation - coarse (total $PM_{>100nm}$) mode.

| Species | Total | Total Aitken | Total | Total coarse | Total $PM_{10}$ |
|---------|-------|--------------|-------|--------------|-----------------|





|  | nucleation mode PM fraction (PM$_{<25nm}$) (%) | mode PM fraction (PM$_{25 - 100\ nm}$) (%) | accumulation mode PM fraction (PM$_{100\ nm-1\ \mu m}$) (%) | mode PM fraction (PM$_{>1\ \mu m}$) (%) | fractional contribution (%) |
|---|---|---|---|---|---|
| SO$_4^{2-}$ | 73.99 | 71.00 | 5.96 | 3.36 | 6.67 |
| NH$_4^+$ | 12.34 | 6.95 | 0.13 | 0.06 | 0.21 |
| Cl$^-$ | 2.36 | 1.98 | 39.96 | 43.36 | 39.54 |
| Na$^+$ | 8.02 | 11.92 | 32.90 | 34.17 | 32.91 |
| MSA | 3.26 | 8.12 | 20.58 | 18.85 | 20.45 |
| HIO$_3$ | 0.004 | 0.004 | 0.05 | 0.05 | 0.05 |
| NO$_3^-$ | 0.006 | 0.01 | 0.17 | 0.15 | 0.17 |

Median mass size distribution and relative mass fraction (Simulation: *BaseCase*)

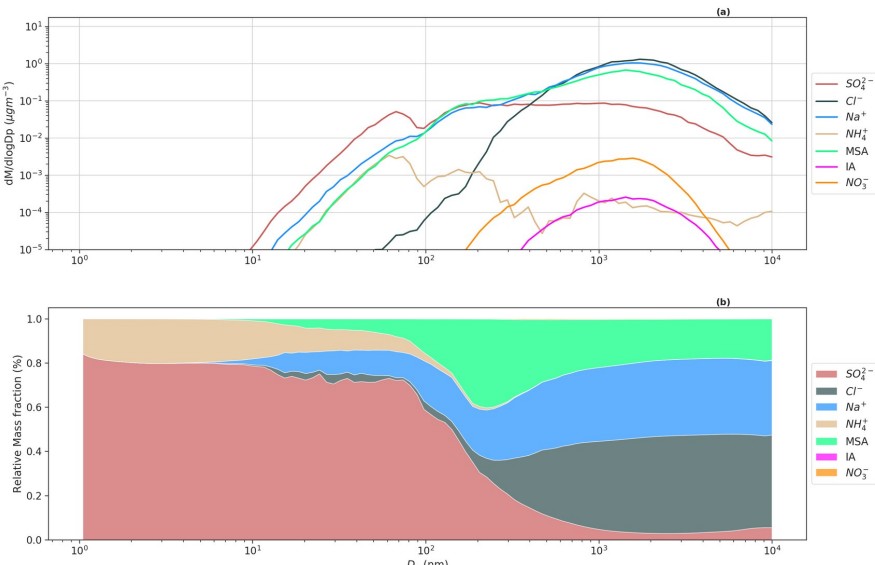

**Figure 5.** Simulated median mass size distribution for *BaseCase* simulations. The upper panel
(a) shows the median mass size distribution for compounds Cl$^-$, Na$^+$, MSA, SO$_4^{2-}$, NH$_4^+$, HIO$_3$
and NO$_3^-$ for the entire size distribution ranging from 1.07 nm-10 μm. The lower panel (b)
shows the relative mass fractions or contribution of compounds Cl$^-$, Na$^+$, MSA, SO$_4^{2-}$, NH$_4^+$,
HIO$_3$ and NO$_3^-$ to total non-refractory PM at different sizes.

Figure 6 compares the daily PM$_{10}$ filter measurements to the modeled values at both





measurement stations. The model prediction of $PM_{10}$ $Cl^-$, $Na^+$, $SO_4^{2-}$ and $NH_4^+$, was evaluated
using statistical metrics such as NMB, FAC2, correlation coefficient ($r$) and RMSE (Table 3).
Though the model does well in simulating the trends of $PM_{10}$ $SO_4^{2-}$, $Na^+$ and $Cl^-$ at Zeppelin ($r$
values of 0.35, 0.51 and 0.6 respectively), it is unable to predict the $NH_4^+$ trends accurately ($r$
= -0.08).
Pearson correlation ($r$ -values) at Gruvebadet are in the range of 0.29-0.34 for $PM_{10}$ $NH_4^+$,
$SO_4^{2-}$, $Na^+$ and $Cl^-$ implying that the model trends are reasonably consistent with the measured
trends. However, at both Gruvebadet the NMB values for $Na^+$ and $Cl^-$ are quite large (1.81
and 1.05), indicating a large overprediction of the predicted values, while $PM_{10}$ $NH_4^+$ and
$SO_4^{2-}$ at Gruvebadet is underpredicted (NMB = -0.88 and -0.28 respectively). In contrast, at
Zeppelin, the modeled PM $SO_4^{2-}$ is overestimated (NMB=1.96). Likewise, large RMSE and
negligible FAC2 values, for $PM_{10}$ $Na^+$, and $Cl^-$ imply discrepancies between the predicted and
measured values, indicating that the model is overestimating $PM_{10}$ $Na^+$ and $Cl^-$ at Gruvebadet
and $PM_{10}$ $SO_4^{2-}$ at Zeppelin. In summary, the model tends to overpredict $PM_{10}$ $Na^+$, $Cl^-$ and
$SO_4^{2-}$ concentrations, but on the other hand, does reasonably well in predicting the daily
measured trends. Additionally, the modeled $PM_{10}$ $Cl^-/Na^+$ molar ratio at Gruvebadet and
Zeppelin is ~0.79 and ~0.95, respectively. This is much higher than the observed $PM_{10}$
$Cl^-/Na^+$ molar ratio at both sites (~0.39). One likely reason for this is the overestimated sea
spray aerosol emissions. The $PM_{10}$ $Cl^-/Na^+$ molar ratios give a measure of the acidic nature of
aerosol, since increased condensation of strong acid MSA and $H_2SO_4$ increases acidity of
aerosols thereby causing loss of $Cl^-$ (dechlorination) as HCl (Ayers et al., 1999; Frey et al.,
2020). Thus, increased availability of $H_2SO_4$ and MSA in particle phase in Aitken mode
particles results in acid-induced $Cl^-$ loss from sea-spray particles.





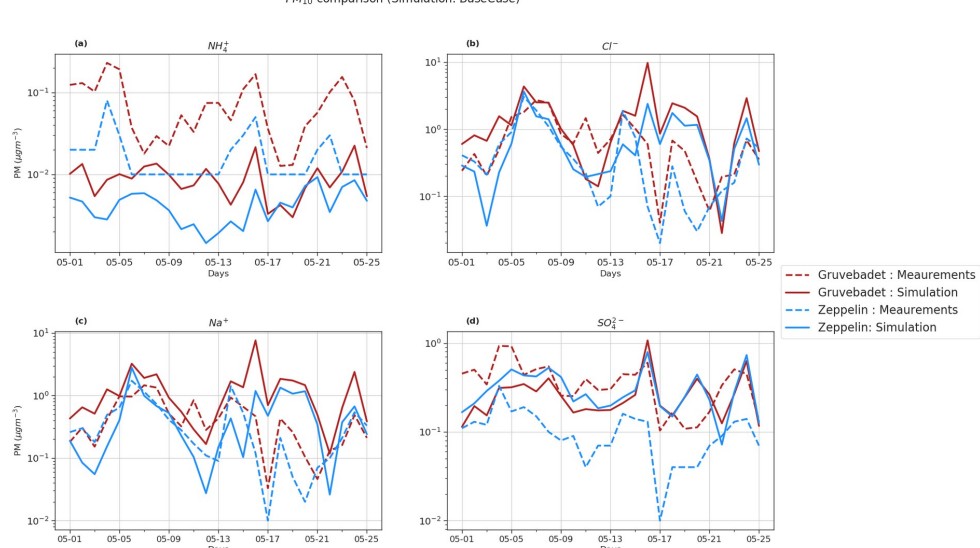


**Figure 6.** $PM_{10}$ comparison of *BaseCase* simulations with daily filter samples from

Gruvebadet and Zeppelin for the entire modeled period. Panel **(a)** shows $PM_{10}$ $NH_4^+$, **(b)**

shows $PM_{10}$ $Cl^-$, **(c)** shows $PM_{10}$ $Na^+$ and **(d)** shows $PM_{10}$ $SO_4^{2-}$ filter samples. The dotted lines

in each panel indicate measurement values, and the solid line denotes simulated values. The

ordinate is plotted in log scale to better visualize the low values.

565

**Table 3**: Evaluation of modeled $PM_{10}$ values at both sites of Gruvebadet (G) and Zeppelin (Z)

for the four particle-phase species $Cl^-$, $Na^+$, $SO_4^{2-}$ and $NH_4^+$.

| Species | Normalized mean bias factor (NMB) | Correlation coefficient (r) | RMSE ($\mu g\ m^{-3}$) | FAC2 |
|---|---|---|---|---|
| $NH_4^+$ | $-0.88^G$, $-0.76^Z$ | $0.34^G$, $-0.08^Z$ | $0.09^G$, $0.02^Z$ | $0.04^G$, $0.2^Z$ |
| $Na^+$ | $1.81^G$, $0.36^Z$ | $0.29^G$, $0.51^Z$ | $1.67^G$, $0.55^Z$ | $0.4^G$, $0.48^Z$ |
| $Cl^-$ | $1.05^G$, $0.39^Z$ | $0.24^G$, $0.60^Z$ | $2.08^G$, $0.74^Z$ | $0.24^G$, $0.44^Z$ |
| $SO_4^{2-}$ | $-0.28^G$, $1.96^Z$ | $0.33^G$, $0.35^Z$ | $0.27^G$, $0.26^Z$ | $0.6^G$, $0.24^Z$ |

568

## 3.3 Vertical profiles of ultra-fine particle

Figure 7 (a) shows the measured vertical $PN_{3-12\ nm}$ concentrations from CPC onboard the UAS

for four measurement periods overlayed onto simulated vertical profiles. Figure 7 (b) and (c)

show the mean vertical profiles for $PN_{3\text{-}12\ nm}$ and $PN_{>12\ nm}$ for both the *BaseCase* simulation and UAS measurements for the entire selected period. The model underestimates the measured $PN_{3\text{-}12\ nm}$ and $PN_{>12\ nm}$ vertical particle number concentrations are underestimated by the model below 200 m a.s.l. The NMB for $PN_{3\text{-}12\ nm}$ and $PN_{>12\ nm}$ is -0.28 and -0.14, respectively, implying that the model underestimates the particle number concentrations. Both the modeled and measured mean particle number concentrations for $PN_{3\ -\ 12nm}$ and $PN_{>12nm}$ are in good agreement between the heights of 200-600 m a.s.l. The lower calculated concentrations of modeled mean particle number concentrations above 600 m a.s.l is most likely affected by higher turbulence in the transition zone from the boundary layer to the free troposphere, which might cause a large mixing of aerosol particles. It should be noted that, at Gruvebadet, the mean SMPS particle number concentrations are in good agreement with the modeled particle number concentrations. However, at the altitude of the Zeppelin station, both the model and UAS measurements of $PN_{>12nm}$ are substantially higher (factor of 4) than the mean particle number concentrations measured with the SMPS at Zeppelin. This finding further strengthens the conclusion that the complex orography at Ny-Ålesund highly affects the variability in the vertical scale, which may cause this discrepancy in the observed and modeled particle number concentrations at Zeppelin (see section 3.1). The UAS measurements were carried out at the airport on Ny-Ålesund (and the UAS was flown around Ny-Ålesund) where the boundary layer measurements, like the model, most likely resemble the general Arctic marine boundary layer conditions. Figure S8 shows the influence of different sensitivity simulations on the modeled vertical particle number concentrations. The large spread in the modeled vertical particle number concentrations in Figure S8, highlights the importance of constraining uncertain parameters such as cloud supersaturation and $NH_3$ gas emissions, to better simulate secondary aerosol formation in marine polar regions.





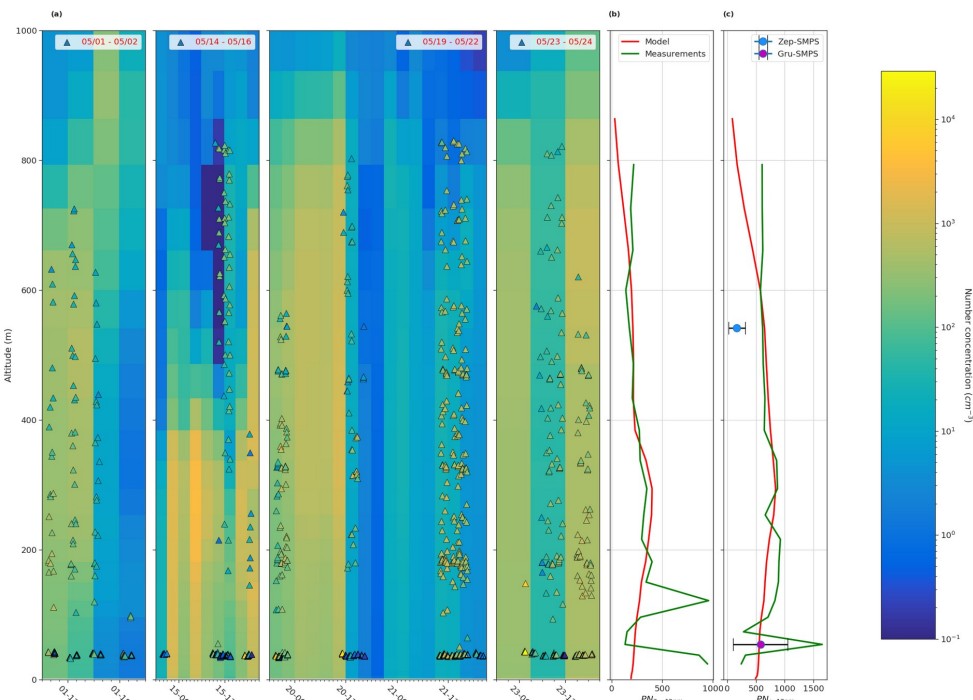

**Figure 7.** Comparison of vertical profiles of measured particle number concentration and
*BaseCase* simulation. Panel (a) shows measured particle number concentration between 3 -12
nm (PN$_{3\text{-}12\,nm}$, triangles), from CPC onboard the UAS during 4 periods 05/01- 05/02, 05/14 -
05/16, 05/19- 05/22, and 05/23- 05/24 (in legend) overlayed onto the simulated PN$_{3\text{-}12nm}$ for
the same periods. Panel (b) shows the simulated and measured mean PN$_{3-12nm}$ and panel (c)
show the PN$_{>12nm}$ for the selected period. Additionally, Panel (c) also shows the mean SMPS
particle concentrations at both Gruvebadet and Zeppelin. The horizontal bars for the mean
SMPS values represent the standard deviation.

## 3.4 Sensitivity Tests

In this section, we will discuss the results from the sensitivity tests that we performed to
complement the main *BaseCase* simulations.

**Median particle size distribution for sensitivity tests**

The sensitivity study *Cloudoff* was performed to test how in-cloud processing affects the



formation of larger particles, especially the accumulation mode (Figure 8). In the *Cloudoff*
test, in-cloud processing was switched off in the model and the RH was set to just below
supersaturation (99.9999%) in the model grid cell where clouds (RH=100.5%) exists in the
*BaseCase* runs. The aim of the *Cloudoff* simulation was to investigate if the model can
capture the observed accumulation mode without aerosol cloud processing. It is clear from
Figure 8 (a) and (b) that in *Cloudoff* simulations, the median size distribution lacks the
accumulation mode and Hoppel minima and has a higher Aitken mode particle concentration
compared to either *BaseCase* or the measured median size distribution. This further
emphasizes the importance of in-cloud processing in activation of particles to CCN sizes and
their growth to larger sizes. Another noteworthy point in *Cloudoff* simulations is the larger
number concentration of particles <10 nm compared to other cases. One plausible reason is
the lack of activated cloud droplets, since the large surface areas of activated droplets are
efficient at Brownian scavenging of smaller particles (Hudson et al., 2015). Likewise, the
median particle number size distribution from the sensitivity tests with lower cloud
supersaturation ($S$) of 0.2% *SSat=0.2,* reduces the accumulation mode particles, since there
are fewer particles with $S_c < S$ available for activation. Increasing $S$ to 0.8% increases
accumulation mode particles, since more particles with $S_c < S$ are activated to cloud droplets
(Aitken mode concentration decreases with respect to *BaseCase* simulations, since more
smaller particles are activated into cloud droplets). Therefore, simulated results show that
increasing the cloud supersaturation results in a higher number of smaller particles being
activated into cloud droplets and shifts the simulated Hoppel minima close to the measured
sizes. Figure S5, in supplementary shows median particle size distribution for all sensitivity
tests.
The *SalterSSA* sensitivity test underestimates both the Aitken and accumulation mode
concentrations at Gruvebadet (Figure S5, supplementary). The Salter sea-spray
parameterization produces ~ 2 magnitudes fewer Aitken mode particles compared to Sofiev
et. al, 2011, while the coarse mode particle emissions using *SalterSSA* parameterization are
higher than Sofiev et. al 2011. This can cause MSA, $H_2SO_4$ and $NH_3$ to partition onto coarse
mode particles rather than contributing to NPF and growth of the nucleation and Aitken mode
particles, which substantially lowers the Aitken and accumulation mode number
concentrations. The *NPFoff* simulation from Figure S5 shows lower Aitken mode
concentrations, implying that the main contributor to Aitken mode particle number
concentrations are the secondary aerosols rather than the primary sea-salt particles.



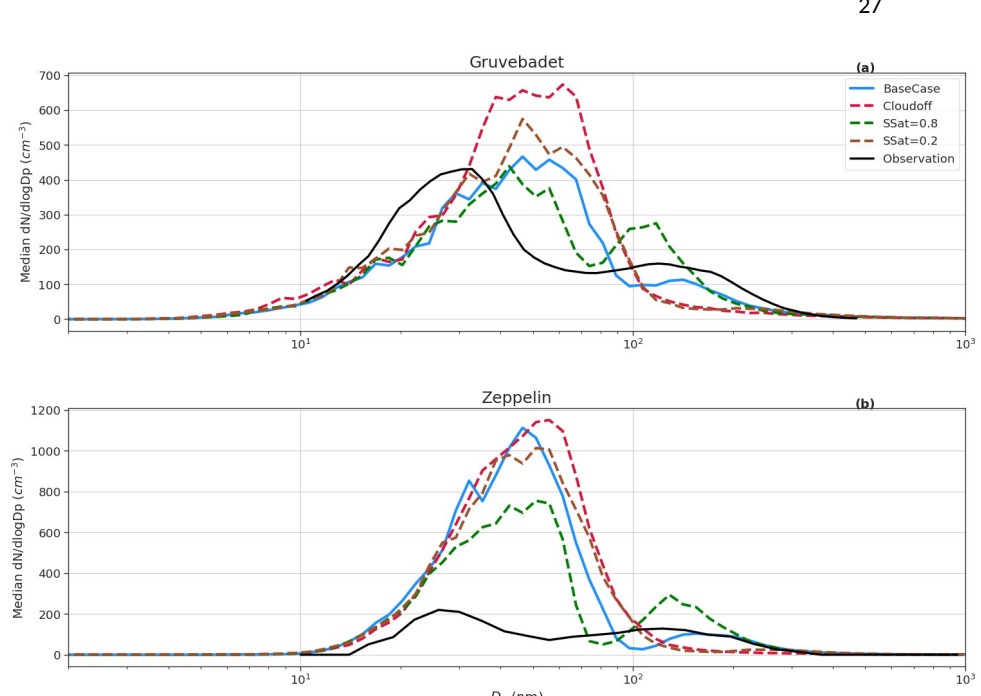

**Figure 8**: Median size distribution at Gruvebadet (panel (a)) and Zeppelin (panel (b)) for all
the sensitivity tests *Cloudoff*, *SSat=0.8*, and *SSat=0.2* (colored dashed lines) including
*BaseCase* (blue solid line) and observations (black solid line).

Another parameter of uncertainty is the concentration of $NH_3$ in the marine atmosphere. The
*LowNH₃* simulations, as expected, result in lower Aitken mode particles, whereas *HighNH₃*
simulations show an overprediction of Aitken mode concentrations (Figure S5,
supplementary). This underlines the necessity of constraining ocean and marine emissions of
$NH_3$ to better predict the aerosol particle formation in marine polar environments.

**Particle phase comparison for sensitivity tests**

Figure 9 shows the contribution of constituent compounds to PM at different particle sizes
with respect to the *BaseCase* simulation. The overall mean contribution of $SO_4^{2-}$ and MSA to
total $PM_{10}$ decreased by ~8% and 11% respectively, in *Cloudoff* runs compared to the
*BaseCase* simulations. It is expected that in non-cloud conditions there is a reduction in $SO_4^{2-}$
and MSA PM contribution because of the reduced partitioning of gaseous $SO_2$ to the cloud
droplets (for PM $SO_4^{2-}$ formation) and inhibition of MSIA ozonolysis in the cloud droplets
(leading to PM MSA formation) (Chen et al., 2018; Hoffmann et al., 2016; Wollesen de Jonge
et al., 2021). This is observed for accumulation mode particles between size ranges of 100 nm



to 1 μm which is characterized by lower $SO_4^{2-}$ and MSA PM. On the other hand, PM $SO_4^{2-}$ and
MSA increase for coarse mode particles (> 1 μm). Without cloud droplet activation the
deliquescent sea spray coarse mode particles become a major liquid water reservoir where
MSIA and to a lesser extent $SO_2$ are dissolved and oxidized into MSA and $SO_4^{2-}$, which partly
explains the increase in PM MSA and $SO_4^{2-}$ for sizes > 1 μm. The results from *Cloudoff*
simulation agrees with the findings from Wollesen de Jonge et al., 2021, who found that MSA
was almost exclusively formed in the aqueous phase via MSIA ozonolysis in cloud droplets
and deliquescent particles during and in between in-cloud periods. PM $SO_4^{2-}$ in *Cloudoff* runs
is mainly driven via condensation of $H_2SO_4$, since an increase in $SO_2$ gas-phase
concentrations (~42% with respect to *BaseCase*) promoted gas-phase $H_2SO_4$ production
(increase of ~44% with respect to *BaseCase*), and therefore $H_2SO_4$ derived PM $SO_4^{2-}$.
In the *woDissolution* simulation, all the PM MSA and $SO_4^{2-}$ are a result of the condensation of
$MSA_{(g)}$ and $H_2SO_{4(g)}$, since irreversible aqueous-phase chemistry is switched off. The overall
contribution of PM $SO_4^{2-}$ to the total $PM_{10}$ increases by  ~12% relative to the *BaseCase* run,
while on the other hand, the contribution of PM MSA decreases by ~87%  (relative to
*BaseCase*). The lower $PM_{10}$ MSA in *woDissolution* simulation emphasizes the importance of
aqueous-phase formation of MSA to the growth of particles. The effect of precipitation on
modeled PM (*NoPrecip*) indicates an increase in PM $Na^+$ and $Cl^-$ of ~112% and 119%
respectively, as compared to *BaseCase* (Figure S7). This is because of the decrease in the wet
deposition of aerosol and sea-spray particles by rain events and below cloud scavenging. The
consequence of neglecting precipitation results in increased condensation sink for $H_2SO_4$ and
$NH_3$ (increase of 62% and 22% in PM $SO_4^{2-}$, $NH_4^+$ respectively), but since sea-spray aerosols
are not scavenged by the wet removal process, the overall fractional contribution to PM by
$SO_4^{2-}$, $NH_4^+$ and MSA is lower relative to *BaseCase* runs.
*SalterSSA* simulation results in higher PM $Cl^-$ and $Na^+$ (470% and 371% increase respectively)
compared to *BaseCase* runs. This is because Salter15 SSA parameterization produces larger
mass emission fluxes in size ranges > 1 μm compared to Sofiev11 SSA parameterization
(Barthel et al., 2019). Additionally, there is an increase of ~19% in PM MSA, largely due to
formation of MSA in larger deliquescent coarse mode particles.



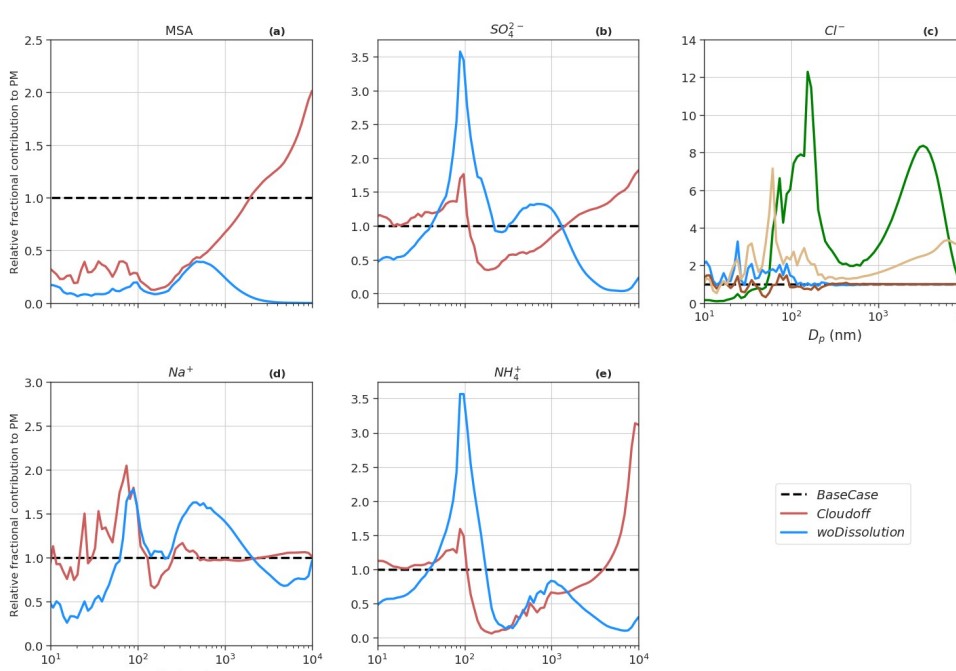

**Figure 9:** Contribution of constituent compounds, namely, MSA (panel (a)), $SO_4^{2-}$ (panel (b)), $Cl^-$ (panel(c)), $Na^+$ (panel (d)) and $NH_4^+$ (panel (e)) to PM with respect to *BaseCase* (the black dotted line).

## 4. Summary and conclusions

In this work, we attempt to simulate secondary aerosol formation at remote Arctic sites of Gruvebadat and Zeppelin, Ny-Ålesund, during the period of 1st - 25th of May 2018. We used the 1-dimensional column model ADCHEM which was run along FLEXPART generated Lagrangian trajectories. Since the air mass spend most of their time over the open ocean upwind of Ny-Ålesund, we use a comprehensive multi-phase DMS chemistry scheme coupled with MCMv3.3.1 and PRAM.

In the model, new particles are formed via ion-mediated $H_2SO_4$-$NH_3$ nucleation, with the initial particle growth mainly driven by condensation of $H_2SO_4$, while the secondary $PM_{10}$ MSA and $SO_4^{2-}$ contribution was mainly formed by oxidation of MSIA and $SO_2$ in the aqueous phase. At Gruvebadet, the modeled median particle number size distribution agrees reasonably well with the measurements, however, at Zeppelin, the simulated Aitken mode median concentration is overestimated by a factor of 5.5. This relatively large discrepancy in



modeled and measured particle size distributions at Zeppelin, and likewise the large
difference between the measured particle number size distributions at Gruvebadet and
Zeppelin, can to a large extent be explained by the orographic effects at Zeppelin which
distorts the atmospheric boundary layer dynamics. Thus, while the model generally is able to
capture the particle number size distribution dynamics in the marine boundary layer, as
measured at the near sea level Gruvebadet site, it generally cannot capture the observations at
the mountain station of Zeppelin, which often lies above the boundary layer and may
experience free tropospheric conditions. This is also supported by the fact that $PN_{>12\,nm}$
concentrations measured with the UAS above Ny-Ålesund airport agrees well with the
modeled particle number concentrations, at the same altitude as Zeppelin. However, both the
model and UAS $PN_{>12nm}$ concentrations is a factor of 4 higher than the $PN_{>12nm}$ observation at
Zeppelin.
Both the measured and modeled particle size distribution, at both stations, show a distinct
Hoppel minima, which can be explained by tín-cloud processing. Model sensitivity runs with
varying cloud supersaturation indicate that a cloud supersaturation of 0.5% or higher is
required for the model to capture the observed Hoppel minima. Furthermore, model
sensitivity runs show that the Aitken mode particle number concentrations are dominated by
contribution of secondary aerosols rather than primary emissions. The modeled $PM_{10}$ $Cl^-$ and
$Na^+$ is positively correlated when compared to $PM_{10}$ filter samples. The main driver for
secondary aerosol particle growth is the formation of MSA via aqueous phase ozonolysis of
the DMS oxidation product MSIA. This demonstrates the importance of multi-phase DMS
chemistry in capturing the size resolved secondary aerosol growth in marine polar regions.
The sensitivity studies indicate that it is important to limit the uncertainties in parameters such
as cloud supersaturation and $NH_3$ emissions over open oceans to get a better constraint on
secondary aerosol formation and its subsequent climatic effects. This work was a first attempt
to simulate new particle and secondary aerosol formation in marine polar regions using a
process based chemistry transport model that includes a comprehensive multi-phase DMS and
halogen chemistry mechanism, detailed gas-molecular cluster and aerosol dynamics. In future
studies, we aim to implement ADCHEM for extended studies in polar marine and remote
continental regions where different atmospheric constituents such as $HIO_3$, terpenes and
amines drive secondary aerosol formation.
**Acknowledgments**
The ALADINA study was funded by the German Research Foundation under grants LA

2907/5-3 and WI 1449/22-3. M. Sipilä acknowledges Academy of Finland (296628) and the European Research Council (ERC) under the European Union's Horizon 2020 research and innovation programme (GASPARCON, grant agreement no. 714621). This project has received funding from the Swedish Research Council Formas project no. 2018-01745-COBACCA, Swedish Research Council VR project no. 2019-05006, the Crafoord foundation through project number 20210969. The presented research has been also been funded by the Academy of Finland (Center of Excellence in Atmospheric Sciences) grant no. 4100200. The authors would like to thank Tinja Olenius from the Swedish Meteorological and Hydrological (SMHI) for help with the implementation of ACDC in ADCHEM. We would also like to acknowledge the invaluable contribution of computational resources from CSC-IT Center for Science, Finland. The authors would like to thank Noora Hyytinen from the University of Oulu and University of Eastern Finland for providing the Henry's law coefficient and dissolution constants which were used in the multi-phase chemistry.

## Author Contributions

CX, PR, MBoy, BA and BW planned and designed the study. PR, RWdJ and CX developed and setup the ADCHEM model. CX, MB and VV performed the FLEXPART model simulations. BA, BW, RoTh, and RT provided the measurement data. Resources were provided by PR and MBoy. CX, PR and MBoy wrote the original draft, which included visualizations made by CX and PR. All other authors discussed the results and contributed to the final manuscript.

## Competing interests

The authors declare that they do have no conflict of interest.

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
