# Peer review of "Secondary aerosol formation in marine Arctic environments: A model measurement"

_Atmospheric Chemistry and Physics, 2022_

## Author Response (AR1)

**Comments for the Reviewers**

We thank the reviewers for positive and constructive comments. The authors response is in blue, the updated text is in red and the omitted text is indicated by the strikethrough.

In addition to the reviewer comments we have also added 3 co-authors Radovan Krejci, Peter Tunved and Mauro Mazzola, who contributed with aerosol particle observations on Svalbard, and one additional affiliation for Michael Boy.

After the submission of the first manuscript version, we noticed and error in the implemented Köhler theory parameterization that describe the activation of the aerosol particles into cloud droplets at the assigned constant cloud supersaturation ($S$) in ADCHEM. The error was that we had mixed up radius and diameter in the curvature and solute term. In practice this mean that when we thought that we were running the model with S=0.5 % it was in fact S=0.25 %. Similar for the sensitivity runs that were expected to represent conditions with S=0.2 % and S=0.8 % the actual S was 0.1 % and 0.4 % respectively. We have illustrated this in figure 1 below. Throughout the manuscript we have changed S=0.5 % to S=0.25 %, S=0.8 % to S=0.4 % and S=0.2% to S=0.1%.

[Figure]

Figure 1. Calculated water vapor supersaturation above the particle surface ($S_c$, calculated using Köhler theory). The blue line illustrates the erroneously implemented Köhler theory parameterizations for pure $(NH_4)_2SO_4$ particles and the red line the new corrected parameterization. In practice the ADCHEM model simulations with the old erroneously implemented Köhler

parameterization using S=0.2 %, S=0.5 % and S=0.8 % represent conditions with S=0.1 %, S=0.25 % and S=0.4 % respectively, as illustrated with the solid vertical lines between the old (wrong) parameterization and the new corrected parameterization.

In addition, we have added a new figure S10 to the supplementary material which illustrate the approximate minimum activation diameter that the different S-values in the base case and sensitivity runs correspond to. We refer to Figure S10 in the updated manuscript on line 252, 290 and line 483 respectively.

[Figure]

Figure S10. Calculated water vapor supersaturation above the particle surface ($S_c$) for pure ammonium sulfate particles at 273.15 K. The cloud supersaturation ($S$) of 0.1%, 0.25% and 0.4% used in the different ADCHEM model simulations are illustrated by the bold black horizontal lines.

**Reviewer 1**

The paper named Secondary aerosol formation in marine Arctic environments: A model measurement comparison at Ny-Ålesund made for an interesting read including a detailed size resolved aerosol model and a good dataset.

Thank you

My impression is however that sometimes the authors have a tendency to explain everything based on the novelty of the approach compared to more traditional works.

RC. The most serious concern to me possibly undermining some of the conclusions is that although the aerosol model is very complex with respect to MSA formation and aqueous formation as well as NH3 related new particle formation (NPF) it seems to lack a number of other basic reactions. It is possible that this information can be found in the referenced material, but if so I think this information can be included easily without. This includes: Aqueous phase production of SO4 from SO2 , e.g. Seinfeld and Pandis (1997) (H2O2 and O3). E.g The last lines of the abstract point to MSA only for the particle growth.

Reply: The first part of my response is in relation to the aqueous-phase production of $SO_4^{2-}$ from $SO_{2(aq)}$, $H_2O_{2(aq)}$ and $O_{3(aq)}$. The aqueous-phase reactions are comprehensive and take into account the existing knowledge pertaining to DMS multi-phase oxidation. The reactions leading to the production of $SO_4^{2-}$ can easily be found in the supplement of the referenced paper by Wollesen and Jonge, 2021. For example formation of $SO_4^{2-}$ by $O_{3(aq)}$ and $SO_2$ is described in equations 727-731 and 744 respectively. To make it easy for the readers, we have added the line L129-130, referring the readers to the paper and supplement by Wollesen and Jonge, (2021).

The line added:

For more details on the DMS, $SO_2$, and halogen multi-phase chemistry scheme used in ADCHEM the reader is referred to the article and supplement of Wollesen de Jonge et al., (2021).

We feel that including the reactions in the current manuscript would make it too cumbersome to the reader since the current work is not related to the development of the scheme, but rather its implementation in the marine boundary layer. As shown in *woDissolution* sensitivity runs MSA which is formed in aqueous-phase is important for the growth of particles, while MSA formed in gas-phase contributes a relatively small fraction to the growth. But alongside aqueous-phase production of MSA, the condensation of $H_2SO_4$ also contributes to the growth of particles but mostly for particles in the Aitken mode, whereas MSA contributes mainly to particles in accumulation mode (c.f. Figure 5 main manuscript).

RC. Gas phase production of SO2 from DMS (Possibly unclear description in the text)

Reply: The gas-phase production of $SO_2$ from DMS is via the abstraction pathway via the formation of an intermediate $CH_3SO_2$ or $CH_3SO$. This is illustrated in the schematic Figure 2 in (Wollesen de Jonge et al., 2021). We had not included it in the manuscript, but refer the readers to the paper by Wollesen de Jonge et al., 2021, included on L129-130.

RC. NPF from other mechanisms, binary or in combinations with organic low volatile compounds, e.g. Vehkamäki et al. (2002), Paasonen et al. (2010). Are these processes found or assumed to be unimportant.

Reply: In this work we assumed that ion-mediated and neutral $H_2SO_4$ - $NH_3$ clusters participate in NPF. Our assumptions agree with the conclusions made by both (Beck et al., 2021; Lee et al., 2020) which show that in spring time Ny-Ålesund enhanced DMS emissions (subsequently $H_2SO_4$) and $NH_3$ are associated with NPF events (Beck et. al 2021 had made observations in May 2017). In summertime highly oxygenated molecules (HOM) can play a crucial role in NPF, but the evidence for NPF at Ny-Ålesund in spring time support $H_2SO_4 – NH_3$ NPF, with very low HOM and iodic acid ($HIO_3$) contribution to NPF. Therefore, we assume that other NPF mechanisms can be ignored, at-least in spring time Ny-Ålesund. This has been mentioned in the manuscript on lines L411-413.

Beck et al (2021) observed dominant contribution of negatively charged $H_2SO_4$ -$NH_3$ clusters to secondary particle formation in May 2017 at Ny-Ålesund, with $HIO_3$ playing a small role in the initial particle formation.

RC. As these are the "traditional" explanations for the reaction pathways and growth I think it is needed to show the relative fraction of these processes compared to new. SO I would like to see the fraction of DMS going to as opposed to SO2, and aqueous phase SO2-SO4 as opposed to DMS-MSA(aq).

Reply: To answer this question, I would like to refer to Table 4 by Wollesen de Jonge et al., (2021), wherein they showed the DMS conversion yield (fractions) to $SO_{2(aq)}$, $SO_4^{2-}$, and MSA for the applied DMS multi-phase chemistry mechanism using a box-model. We can consider only 2 cases, AtmMain (analogous to *BaseCase* in current manuscript, where all in-cloud processing and irreversible aqueous-phase chemistry is considered) and woAqAtm (without irreversible aqueous-phase chemistry, analogous to *woDissolution* case).

I assume by "traditional" pathways the reviewer refers to case without multi-phase chemistry (wo AqAtm). Since we only save the modelled concentrations and not the actual reaction pathways, deposition loss rates and the transport flux in and out of the model grid cells we cannot fully quantify the various fractions of DMS which is oxidized to $SO_2$, HPMTF, $SO_4^{2-}$ and MSA in the current work. But based on the work by Wollesen de Jonge et al., (2021), we can infer the trends of different pathways. As seen from Table 4 (in Wollesen de Jonge et al., (2021),) if we consider the

traditional pathway (woAqAtm) $SO_4^{2-}$ is the major contributor to particle mass (PM), due to condensation of $H_2SO_4$, while the contribution of MSA to PM is reduced, since MSA production is dominated in aqueous-phase. In woAqAtm gas-phase $SO_2$ concentrations increase since $SO_{2(g)}$ is not transitioning to the aqueous-phase. When aqueous-phase is considered (AtmMain) $[SO_{2(g)}]$ is reduced (due to partitioning to aqueous-phase), while MSA and $SO_4^{2-}$ PM contribution increases. The increasing $SO_4^{2-}$ contribution, in comparison to traditional case is due to $SO_2$ being oxidized to $SO_4^{2-}$ by $O_3$ and $H_2O_2$ in the aqueous-phase.

RC. I have also a general question about the supersaturation experiments.
"Increasing S to 0.8% increases accumulation mode particles, since more particles with Sc < S are activated to cloud droplets (Aitken mode concentration decreases with respect to BaseCase simulations, since more smaller particles are activated into cloud droplets)."
Even for S of 0.8 % the critical diameter is more than 50 nm, actually. I think the observed local number minimum may work as a proxy for activation size.
Reply: Good point. We agree with the reviewer that the Hoppel minimum can be used as a proxy for the activation size. We will consider this approach in our future work. For S = 0.4 % (erroneously described as 0.8 % in the submitted manuscript) the minimum diameter of activation into cloud droplets are ~ 60 nm for the sulfate rich particles with water uptake similar as ammonium sulfate (see the new Figure S10).

Figure 8 however show an impact already at 20-30 nm. Is it possible that the reason for the reduction is not from direct activation of the particles but rather an increased cloud droplet surface area, i.e the inverse of the explanation given for cloudoff.
Reply: Thank you for this comment. We agree with the reviewer that we cannot just attribute the reduction in particle number concentration to activation of particles to cloud droplets since this reduction in Aitken mode particle could also result from the Brownian scavenging by large cloud droplets. We have added a sentence on L610 in the manuscript to make this point clear.
The new text is:
It should be noted, that other processes such as coagulation scavenging by larger cloud droplets also contribute to the shift in particles from the Aitken mode to accumulation mode (as seen in median measured particle number size distributions, Figure 8, Noble and Hudson, (2019)).

RC. Figure S8 shows as far as I can see the same lines for NPFOff and no dissolution. Does NPF in the model depend on cloud processes?

Reply: Thank you for pointing this out. That was a small error in plotting where I mixed up the data. We have updated the Figure S9 (earlier Figure S8), and as shown below in Figure C1 *NPFoff* and *woDissolution* vertical profiles for $N_{3\text{-}12\,nm}$ and $N_{>12\,nm}$ are different. The NPF in the model does not depend on cloud processes. The NPF in the model is based solely on the ACDC code, which in this study is based on ion-mediated and neutral $H_2SO_4$-$NH_3$ clustering. During the UAS flight campaign the vertical profiles of *NPFoff* and *woDissolution* case for $N_{3\text{-}12nm}$ case look similar, but the vertical profile for $N_{>12nm}$ are completely different, with the *woDissolution* case predicting much larger $N_{>12nm}$ vertical concentrations. The smaller $N_{3\text{-}12nm}$ concentrations is most likely due to increased uptake of gas-phase $H_2SO_4$ by condensation onto larger particles, thereby inhibiting NPF.

[Figure]

**Figure C1:** *NPFoff* and *woDissolution* case for $N_{3\text{-}12\,nm}$ and $N_{>12\,nm}$.

RC. One last question question with respect to assumptions about the relatively high number of 3-12 nm shown in observations figure S8, compared to the apparently lower number found in figure 8. It is hard to compare visually with log-normal distributions, but still figure 8 does seem to give a much lower number of for the 3-12 nm size.

Reply: Thank you for this question. This confusion could arise due to 2 reasons:

1. Only SMPS data starting from 10 nm was used in Figure 8 to show the median particle size distributions and no observational data (UAS data) is presented in the 3 – 10 nm ranges. To overcome this confusion, we have updated the Figure 8 to show particle diameters in range between 10 nm – 1μm.

2. Figure 7 shows the mean $N_{3-12\ nm}$ concentration while Figure 8 shows the median size distribution. In case we plot the mean particle number size distribution for the UAS measurement period (as shown in Figure C2) we see that the $N_{3-12\ nm}$ concentrations are higher and are comparable to the simulated vertical profiles shown in Figure 7. We can add this to the supplement if the reviewer thinks its necessary.

[Figure]

**Figure C2**: Mean particle size distribution during the UAS measurement period.

**General comment:**

RC. It is easy for the reader to mix up PM and PN. I think it is common to use only N for the number but that is more of a suggestion and I leave that to the authors.

Reply: We understand that readers might get confused, hence we have changed PN → $N$ in the text and in figure 7 and supplementary Figure S8.

RC. Last section of abstract, as discussed above. Are all processes included?

Reply: Yes, in the last section of the abstract we discuss results from the *BaseCase* simulation which included all the processes.

**Details:**

--"size < 12nm" Please define size as radius or diameter the first time you define it. Later on size is fine as long as it retains the same definition.

Reply: Thank you for pointing this out. This has been changed now to "particle diameters <12 nm" on line L 155.

RC. Measurement period. Did you experiment with classifying the trajectories when discussing the results. Eg. a western airflow is expected to have both marine and more clouds than a easterly flow so the trajectories with marine characteristics may have experienced lower emissions than continental pathways so the size distribution may also be caused by different emissions, not only the cloud processing.

Reply: Interesting question. Yes in fact we have classified the trajectories with respect to the measurement station. Figure C3 shows all the trajectories arriving at Zeppelin during the study period and table T1 indicates the fraction of air-mass from different quadrants namely North-West (N-W), South-West (S-W), North-East (N-E) and South-East (S-E) with respect to the measurement site (Zeppelin). Since the emissions are read in along the trajectories, and most of the trajectories are westerly (~72 %) the change in emission source strength between easterly and westerly air-masses will be accounted in the model. The *BaseCase* and *CloudOff* cases are run along the same trajectories and emission sources, hence any change in size distribution, for e.g. Hoppel minimum are entirely a result of in-cloud processing.

Table T1: The table shows the fraction of air-mass arriving at Zeppelin form the four quadrants.

| Region | Fraction of air mass (~%) |
| --- | --- |
| North-West | 8 % |
| South-West | 64 % |
| North-East | 6 % |
| South-East | 22 % |

Figure C4 shows the different airmasses (eastern or western) as we can clearly see the Hoppel minimum regardless of the airmass direction. Based on this we can safely conclude that the in-cloud processing is the main factor contributing to the Hoppel minimum.

[Figure]

**Figure C3**: Trajectories arriving at Zeppelin during the measurement period of 1$^{st}$-25$^{th}$ May 2018. The black diamond represents the measurement site.

[Figure]

**Figure C4** : The median size distribution for different air masses (eastern or western).

RC. Section 2.2 Sea surface temperature above 0. --> No trajectories from areas with sea-ice so no negative SST?

Reply: The SST in this context refers to temperatures only over the open ocean. The minimum SST is n this study is -2 °C and the maximum SST is 23.05 °C. The SST is used to along with wind speed at 10m to estimate the sea-spray aerosol, therefore we only consider SST over open ocean.

RC. Table 1: Please make the table smaller and more readable. Also I think it is useful for the reader if you also refer to the table in the results section.

Reply: Done. We added a line to direct the readers to the Table 1. The new text is:

In this section, we will discuss the results from the sensitivity tests that we performed to complement the main *BaseCase* simulations. The settings of different sensitivity tests are described in Table 1.

RC. Table 3. For readability please consider using the same order of species in the text as in the table.

Reply: Done. We have modified the text and table 3 (in the preceding paragraph) as follows:

Pearson correlation ($r$ -values) at Gruvebadet are in the range of 0.29-0.34 for $PM_{10}$ $NH_4^+$, $SO_4^{2-}$, $Na^+$ and $Cl^-$ implying that the model trends are reasonably consistent with the measured trends. However, at Gruvebadet the NMB values for $PM_{10}$ $NH_4^+$ and $SO_4^{2-}$ are underpredicted (NMB = -0.88 and -0.28 respectively), while $PM_{10}$ $Na^+$ and $Cl^-$ show a large overprediction (1.81 and 1.05) in the modelled values. In contrast, at Zeppelin, the modeled PM $SO_4^{2-}$ is overestimated (NMB=1.96). Likewise, large RMSE and negligible FAC2 values, for $PM_{10}$ $Na^+$, and $Cl^-$ imply discrepancies between the predicted and measured values, indicating that the model is overestimating $PM_{10}$ $SO_4^{2-}$, $Na^+$ and $Cl^-$ at Gruvebadet and $PM_{10}$ $SO_4^{2-}$ at Zeppelin.

| Species | Normalized mean bias factor (NMB) | Correlation coefficient (r) | RMSE (µg m$^{-3}$) | FAC2 |
|---------|-----------------------------------|-----------------------------|--------------------|------|
| $NH_4^+$ | -0.88$^G$ , -0.76$^Z$ | 0.34$^G$, -0.08$^Z$ | 0.09$^G$, 0.02$^Z$ | 0.04$^G$, 0.2$^Z$ |
| $SO_4^{2-}$ | -0.28$^G$, 1.96$^Z$ | 0.33$^G$, 0.35$^Z$ | 0.27$^G$, 0.26$^Z$ | 0.6$^G$, 0.24$^Z$ |

| | | | | |
|---|---|---|---|---|
| Na$^+$ | 1.81$^G$, 0.36$^Z$ | 0.29$^G$, 0.51$^Z$ | 1.67$^G$, 0.55$^Z$ | 0.4$^G$, 0.48$^Z$ |
| Cl$^-$ | 1.05$^G$, 0.39$^Z$ | 0.24$^G$, 0.60$^Z$ | 2.08$^G$,0.74$^Z$ | 0.24$^G$, 0.44$^Z$ |

RC. line 678. Sea-spray aerosols are not scavenged --> Why?

Reply: In the *NoPrecip* case the wet deposition is switched off implying that the rain events and below cloud scavenging of aerosol particles including sea-spray aerosols are inhibited. This results in sea-spray aerosols not scavenged by wet deposition.

RC. 721: Typo tin-cloud --> in-cloud

Reply: Done.

The manuscript describes a modelling study using the ADCHEM model updated with a complex MSA-halogen mechanism in simulating new particle formation in the Arctic, and the comparison using observation from Arctic sites and campaigns. The manuscript is very-well written and easy to follow, and the results nicely support the conclusions. I find the manuscript suitable for publication in ACP, after addressing the comments and corrections I have listed below.

Thank you

**Introduction**

RC. What is the aim of the study and the hypothesis? A paragraph at the end of the introduction section on this could be useful.

Reply: We have added a few lines introducing the aim of this study to the last paragraph of the introduction section.

The aim of this work is to understand the processes and DMS oxidation products governing the formation and growth of the secondary aerosol in pristine remote marine Arctic region. To facilitate this, we have implemented the above mentioned DMS multi-phase chemistry mechanism into ADCHEM (see Methods section) and modeled the aerosol formation along air mass trajectories arriving at Ny-Ålesund.

RC. Line 57: Recent studies (e.g. Lenssen et al., 2019) suggest the warming rate is up to a factor of three).

Lenssen, N. J. L., Schmidt, G. A., Hansen, J. E., Menne, M. J., Persin, A., Ruedy, R., and Zyss, D.: Improvements in the GIS- TEMP Uncertainty Model, J. Geophys. Res.-Atmos., 124, 6307– 6326 https://doi.org/10.1029/2018jd029522, 2019.

Reply: Thank you for the update. We have now updated the text now and added two more references.

Lenssen, N. J. L., Schmidt, G. A., Hansen, J. E., Menne, M. J., Persin, A., Ruedy, R., and Zyss, D.: Improvements in the GIS- TEMP Uncertainty Model, J. Geophys. Res.-Atmos., 124, 6307– 6326 https://doi.org/10.1029/2018jd029522, 2019.

AMAP: AMAP Arctic Climate Change Update 2021: Key Trends and Impacts, , 16 pp [online] Available from: https://www.amap.no/documents/download/6759/inline, 2021.

The updated text is:

The Arctic environments are susceptible to perturbations in the radiation balance, with some estimates suggesting that, compared to the global average, the Arctic is warming at three times the rate, a phenomenon termed as Arctic amplification (AMAP, 2011, 2017, 2021; Lenssen et al., 2019; Tunved et al., 2013)

RC. Line 72: Correct as Arrigo and van Dijken (2015), and throughout the manuscript.

Reply: Thank you for pointing this out. Corrected.

**Methods**

RC. Line 148: Why is this period chosen? Is it a period of observed NPF or is it the ALANDIA campaign? How does the model behave in non-NPF periods?

Reply: This study period was chosen since the ALADINA campaign was conducted during this period. One reason for selecting this period for the ALADINA campaign could be since the month of May signifies a transition from the winter period (Arctic haze, dominated by accumulation mode particles) to the spring period characterized by snow melt, increase in biological activity and incoming solar radiation, thereby facilitating an onset of NPF in Ny-Ålesund. I assume by non-NPF periods the reviewer implies days during the campaign when no NPF was observed. For example, as shown in Figure 1, the model and observations are good agreement during the period between 16th - 20th May when no NPF events were observed.

RC. Line 159: Are there also PM2.5 chemical composition measurements available?

Reply: Thank you for the question. We used quality controlled data from both measurement stations in this work e.g., from the EBAS portal. Unfortunately, we did not find any quality controlled PM2.5 data from the EBAS portal.

RC. Are biomass burning emissions not taken into account?

Reply: Thank you for the question. No, we did not include any biomass burning emissions in this work, because we assumed that they were not too many forest fires in the month of May.

**Results**

RC. Figures 1 and 2. Is it possible to show a third panel where obs-model is shown? It would be useful for the reader to compare visually what is written in the text.

Reply: Thank you for this suggestion. It is not easy to add an obs-model plot, since the observations and model data (dNdlogdp) are in different diameter bins. However, we have added to Figures 1 and 2 an additional subplot (c) which shows both the measured and simulated total number concentration (cm$^{-3}$). This would make it easier for the readers to compare visually what is written in the text.

The following figures has been added:

**Figure C5.** Particle number size distribution at Gruvebadet for *BaseCase*. The panel **(a)** shows the measurement data for the period 1-25$^{th}$ May from SMPS (10 nm-470 nm) and NAIS (2.5 - 10 nm), the panel **(b)** provides the modeled particle size distribution and panel **(c)** shows the total measured and simulated number concentrations. The black line at 10 nm denotes the boundary above which SMPS data starts and NAIS data ends. The abscissa indicates the time for the entire simulated duration. The ordinate in Figure C5 for both panels (a) and (b) indicates the particle diameter (D$_p$, nm).

[Figure]

**Figure C6.** Particle size distribution at Zeppelin. The panel **(a)** shows the measurement data for the period 1-25$^{th}$ May from SMPS, the panel **(b)** provides the simulated particle size distribution and panel **(c)** shows the total measured and simulated number concentrations for the *BaseCase*

simulations. The abscissa and ordinates are similar to Figure C5.

Line 369: Is it Figure 1a or Figure 2a?

Reply: Thank you for pointing this out. The reference here is to Figure S2a. This has now been corrected in the manuscript.

Line 449: Does the model take into account the sulfate production via in-cloud oxidation of SO2? The model discrepancy highlighted in 466-471 can be a result of this process not taken into account.

Reply: Thank you for the question. Yes, during in-cloud periods $SO_{2(aq)}$ undergoes subsequent oxidization by $O_{3(aq)}$ and $H_2O_{2(aq)}$ to form $SO_4^{2-}$ PM (Wollesen de Jonge et al., 2021). The discrepancy in Hoppel minimum location is most likely due to the assumed supersaturation values. It could be estimated that the cloud supersaturation could be >0.8 %, which can result in the observed Hoppel minima location of ~60 nm.

Figure 4 could be considered to be removed as the text does not provide any discussion on the temporal variation or magnitude compared to earlier measurements. I would rather provide a figure (a bar plot or box whisker) comparing the simulations with the measurements.

Reply: Thank you for the suggestion. We have updated the Figure 4 with the following Figure C7. It should be noted that the measured values are from a 2017 campaign from Gruvebadet and not from the same year (2018) when the simulations were performed. We have updated the text in section 3.2 to:

Previous text:

~~Figure 4 shows the simulated gas-phase precursor and main DMS oxidation product concentrations including $H_2SO_4$, MSA, MSIA, HPMTF, $SO_2$ and DMSO, for the entire period at the height levels representing Gruvebadet (G1 and G2), and Zeppelin (Z1 and Z2). The $SO_2$ gas-phase concentrations are in the order of $10^6$-$10^9$ # cm$^{-3}$ (with monthly mean values 1.7 x $10^8$ # cm$^{-3}$), which is a factor of 2.3 higher than the average concentrations measured for spring 7.6 x$10^7$ # cm$^{-3}$ by (Lee et al., 2020) at Zeppelin. The monthly mean simulated $H_2SO_4$ gas phase concentrations (6.8 x $10^5$ # cm$^{-3}$) also agree well with the estimated $H_2SO_4$ proxy (Eq. S1, supplementary) spring average values of 7.5 x $10^5$ # cm$^{-3}$ (Lee et al., 2020) at Zeppelin.~~

Updated Text:

Figure 4 shows the range of simulated gas-phase concentrations of DMS oxidation products $H_2SO_4$, MSA and $HIO_3$ for the entire period at height levels representing Gruvebadet. The mean measurement values (red dots) represent gas-phase concentrations for the same species from an earlier 2017 May campaign performed at Gruvebadet by Beck et al., (2021). Measurements of $H_2SO_4$ at Gruvebadet from May 2017 indicate monthly mean concentrations around $\sim10^6$ # $cm^{-3}$ (Beck et al., 2021). The modeled $H_2SO_4$ concentrations at Gruvebadet are 3 x $10^6$ # $cm^{-3}$, implying a reasonably good model performance in predicting gaseous precursor concentrations.

The previous Figure 4 (now → Figure S4) and relevant text (below) has been moved to the supplement:

The $SO_2$ gas-phase concentrations are in the order of $10^6$-$10^9$ # $cm^{-3}$ (with monthly mean values 1.7 x $10^8$ # $cm^{-3}$), which is a factor of 2.3 higher than the average concentrations measured for spring 7.6 x$10^7$ # $cm^{-3}$ by (Lee et al., 2020) at Zeppelin (Figure S4). The monthly mean simulated $H_2SO_4$ gas phase concentrations (6.8 x $10^5$ # $cm^{-3}$) also agree well with the estimated $H_2SO_4$ proxy (Eq. S1, supplementary) spring average values of 7.5 x $10^5$ # $cm^{-3}$ (Lee et al., 2020) at Zeppelin.

[Figure]

**Figure C7**: The figure shows the comparison of three gas phase species, $H_2SO_4$, $HIO_3$ and MSA compared with the mean measured values (red dots) from the campaign conducted one year earlier in
2017 at Gruvebadet in the month of May. Figure C7 shows the modeled values visualized as a box plot. The lower and upper edge of the box plot denote the $25^{th}$ and $75^{th}$ quartile values, while the middle line in each box indicates the median values. The whiskers (lower and upper) indicate the minimum and maximum values.

Figure 5b and Table 2 shows the same information in different ways, which is well described in the text. I would move one of them to the supplement.
Reply: That's a good idea. The table 2 has been moved to the supplement.

Line 573: Correct the sentence.
Reply: Thank you for pointing this. Done. The updated sentence is:
The model underestimates the measured $N_{3\text{-}12\ nm}$ and $N_{>12\ nm}$ vertical particle number concentrations below 200 m a.s.l.

Line 575: Are the NMB values representing under 200 m asl or the whole vertical extent? Can the good agreement in the 200-600 m compared to the first 200 meter imply that the model is doing poorly close the local sources, which can be attributed to the uncertainty in the emissions, while 200-600 meters represent more transported particles and the model captures this transport?
Reply: The NMB values represent the entire vertical extent from the ground to 800 m. We agree with the reviewer that we might be missing sudden changes in wind direction, which can alter local emission source and strengths, which will affect the model performance near the surface and up to 200 m. Above 200 m it seems plausible that the model captures the long range transported particles and emissions, which improves the model predictability.

**References**

Beck, L. J., Sarnela, N., Junninen, H., Hoppe, C. J. M., Garmash, O., Bianchi, F., Riva, M., Rose, C., Peräkylä, O., Wimmer, D., Kausiala, O., Jokinen, T., Ahonen, L., Mikkilä, J., Hakala, J., He, X. C., Kontkanen, J., Wolf, K. K. E., Cappelletti, D., Mazzola, M., Traversi, R., Petroselli, C., Viola, A. P., Vitale, V., Lange, R., Massling, A., Nøjgaard, J. K., Krejci, R., Karlsson, L., Zieger, P., Jang, S., Lee, K., Vakkari, V., Lampilahti, J., Thakur, R. C., Leino, K., Kangasluoma, J., Duplissy, E. M., Siivola, E., Marbouti, M., Tham, Y. J., Saiz-Lopez, A., Petäjä, T., Ehn, M., Worsnop, D. R., Skov, H., Kulmala, M., Kerminen, V. M. and Sipilä, M.: Differing Mechanisms of New Particle Formation at Two Arctic Sites, Geophys. Res. Lett., 48(4), 1–11, doi:10.1029/2020GL091334, 2021.

Lee, H., Lee, K., Lunder, C. R., Krejci, R., Aas, W., Park, J., Park, K.-T., Lee, B. Y., Yoon, Y. J. and Park, K.: Atmospheric new particle formation characteristics in the Arctic as measured at Mount Zeppelin, Svalbard, from 2016 to 2018, Atmos. Chem. Phys., 20(21), 13425–13441, doi:10.5194/acp-20-13425-2020, 2020.

Wollesen de Jonge, R., Elm, J., Rosati, B., Christiansen, S., Hyttinen, N., Lüdemann, D., Bilde, M. and Roldin, P.: Secondary aerosol formation from dimethyl sulfide – improved mechanistic understanding based on smog chamber experiments and modelling, Atmos. Chem. Phys., 1–33, doi:10.5194/acp-2020-1324, 2021.

Wollesen de Jonge, R., Elm, J., Rosati, B., Christiansen, S., Hyttinen, N., Lüdemann, D., Bilde, M. and Roldin, P.: Secondary aerosol formation from dimethyl sulfide – improved mechanistic understanding based on smog chamber experiments and modelling, Atmos. Chem. Phys., 21, 9955–9976, 2021